# Enteric Release Essential Oil Prepared by Co-Spray Drying Methacrylate/Polysaccharides—Influence of Starch Type

**DOI:** 10.3390/pharmaceutics12060571

**Published:** 2020-06-19

**Authors:** Ioannis Partheniadis, Evangelia Zarafidou, Konstantinos E. Litinas, Ioannis Nikolakakis

**Affiliations:** 1Department of Pharmaceutical Technology, School of Pharmacy, Faculty of Health Sciences, Aristotle University of Thessaloniki, 54124 Thessaloniki, Greece; ioanpart@pharm.auth.gr (I.P.); evanzara@pharm.auth.gr (E.Z.); 2Laboratory of Organic Chemistry, Department of Chemistry, Aristotle University of Thessaloniki, 54124 Thessaloniki, Greece; klitinas@chem.auth.gr

**Keywords:** microencapsulation, oregano essential oil, electrostatics, methacrylate polymer, emulsion reconstitution, oral delivery, powder flow

## Abstract

Oregano essential oil (EO) enteric release powder was formulated by spray drying feed emulsions stabilized with polysaccharides (PSC) and Eudragit^®^ L100 (PLM). Different modified starches were used in the PSC component. Spray-dried powders were evaluated for particle size and morphology, dynamic packing, flowability, chemical interactions, reconstitution, and gastric protection. Feed emulsions were stable, indicating the good emulsification ability of the PLM/PSC combination. The presence of polymer in the encapsulating wall neutralized electrostatic charges indicating physical attraction, and FTIR spectra showed peaks of both PLM and PSC without significant shifting. Furthermore, the presence of polymer influenced spray drying, resulting in the elimination of surface cavities and the improvement of powder packing and flowability, which was best when the surface-active, low-viscosity sodium octenyl succinate starch was used (angle of repose 42°). When a PLM/PSC ratio of 80/20 was used in the encapsulating wall, the spray-dried product showed negligible re-emulsification and less than 15% release in pH 1.2 medium for 2 h, confirming gastric protection, whereas at pH 6.8, it provided complete re-emulsification and release. In conclusion, (1) polymer–PSC physical interaction promoted the formation of a smoother particle surface and product with improved technological properties, which is important for further processing, and (2) the gastro protective function of Eudragit^®^ L100 was not impaired due to the absence of significant chemical interactions.

## 1. Introduction

Essential oils are widely exploited as natural antimicrobials due to their accepted safety and efficacy profiles [1,2]. The strong and rapid bacteriostatic and bactericidal activity of oregano essential oil (EO) against Gram-negative and Gram-positive bacteria and in particular *E. coli* (activity within 1 min) is now well-established both by direct addition into diffusion plates inoculated with bacteria culture [3,4] but also by addition to diffusion plates as re-emulsified spray dried powder [5]. Later studies have also demonstrated antiviral activity of EO against a range of human/animal viruses: Gilling et al. [6] and Sánchez et al. [7] demonstrated that EO inactivated murine norovirus (MNV) and other enteric viruses such as the picornaviruses, caliciviruses, astroviruses, and hepatitis by acting directly on the viral capsid; Pilau et al. [8] demonstrated the activity of EO against bovine viral diarrhea virus (BVDV); Choi et al. [9] showed activity against anti-enterovirus. Gaur et al. demonstrated the anti-parasitic action of EO against *Cryptosporidium parvum*, which is the second leading cause of persistent diarrhea among children in low-resource settings [10].

From the above published data, it seems that the EO activity is exerted on pathogens that mainly reside in the intestine, and this has also been demonstrated in vivo: Zou et al. showed that EO-treated pigs had lower *E. coli* population in the jejunum, ileum, and colon due to the promotion of intestinal barrier integrity [11]. Katsoulos et al. showed that the administration of oregano EO in calves prevented neonatal diarrhea syndrome and reduced mortality by inhibiting coliform bacterial overgrowth in the small intestine due to its antibacterial activity against *E. coli* [12]. These results prove that in order to release its full potency, EO should be delivered directly to the intestine. Since its main constituent carvacrol, which is also responsible for the antimicrobial activity, is partly absorbed in the gut after oral administration [13], less amount reaches the intestine, thus reducing the therapeutic efficacy. Therefore, a microencapsulated EO oral formulation is needed to provide gastric protection and enteric release, besides masking the bitter taste and protecting the EO bioactive ingredients during processing, transportation, and storage.

Several attempts have been made to prepare enteric EO formulations. Omonijo et al. (2018) formulated microparticles containing thymol and lauric acid for delivery to pig intestine using starch and alginate via melt-granulation [14]. The alginate significantly decreased the release of thymol and lauric acid in simulated gastric fluid (SIG) and increased their release in simulated intestinal fluid (SIF). Ma et al., formulated trans-cinnamaldehyde (CIN) with an adsorbent powder and fatty acid using a melt-solidification technique and coated them with an enteric polymer [15]. CIN was mostly retained in the simulated gastric fluid but released rapidly (>80% under 2 h) in SIF. Zhang et al., used alginate and whey protein to encapsulate carvacrol [16]. After administration in chickens, they found good gastric resistance and high carvacrol concentrations in the jejunum and ileum. However, a negligible amount was detected in the intestine of chickens fed with non-encapsulated carvacrol.

Spray drying is a manufacturing process that is suitable for the microencapsulation and protection of EO ingredients sensitive to humidity and light [17,18]. It yields spherical particles within a narrow size range and good flowability, which are important attributes for further processing into final dosage forms [5,19]. In addition, spray drying is easily scaled up, since development trials and production batches run on equipment operating on the same principle differing only in capacity. So far, its application for the preparation of enteric release formulations has been restricted to pure drugs [20]. To our knowledge, there is no published work on the formulation of enteric release essential oil by spray drying, providing at the same time a technologically acceptable product with commercial potential.

Therefore, the objective of this work was twofold: first to develop a microencapsulated enteric EO powder formulation by co-spray drying EO polysaccharide dispersions (PSC) with the enteric polymer Eudragit^®^ L100 (PLM). Different starch grades differing in surface activity and viscosity were tried to optimize the technological performance of the product. Previous attempts have shown that the combination of polysaccharides with methacrylic polymer did not affect the pH sensitivity of methacrylic [21]. In a second step, the work was set to evaluate the gastric protecting and enteric release efficiency of the products in the light of physicochemical interactions between the PSC components and the polymer.

## 2. Materials and Methods

### 2.1. Materials

Oregano essential oil of Greek origin (EO, 85.89% carvacrol) from Oreganum vulgare (Heracleoticum) was gifted by Ecopharm Hellas, Kilkis, Greece. Arabic gum (AG, Spraygum AB) was from Nexira, France. Maltodextrin (MD, Glucidex 21), modified starches (MS, Clearam^®^ CH 2020, food grade acetylate di-starch adipate, E1422) and Cleargum^®^ (food grades sodium octenyl succinate starch, E1450 of high CO 01 and low CO 03 viscosity) were all from Roquette Italia, gifted by Interallis Chemicals, Greece. For practical reasons and to link with their chemistry, the starches were coded: Clearam^®^ as ADA; Cleargum^®^ as SOS (SOS1 grade for high and SOS3 for low viscosity). Eudragit^®^ L100 was from Evonik (Darmstadt, Germany) kindly gifted by ChemiX SA, Greece. The chemical structures of the two chemically different modified starches are shown in Figure 1.

### 2.2. Methods

#### 2.2.1. Viscosity of Starch Grades

The viscosity of aqueous solutions of starches at concentrations 3.6% *w*/*w* (as in the feed emulsions) or 7.4% *w*/*w* and 10.0% *w*/*w* was measured with a rotational viscometer (HAAKE Viscotester VT24, Thermo Fisher Scientific, Karlsruhe, Germany) fitted with a cup-and-bob NV ST sensor. First, 20 mL were added in the gap between the cup and bob, and the torque was read on the scale at increasing rotation speeds from 2.8 to 22.6 rpm corresponding to shear rates of 15.3 to 122 s^−1^. Shear stress, σ (Pa), and shear rate, $\dot{\gamma }$ (s^−1^), were obtained using Equations (1) and (2) and σ versus $\dot{\gamma }$ graphs were plotted.
σ (Pa) = 0.029 × Scale reading(1)
$$\dot{\gamma }\;(s^{-1})=122/\mathrm{speed setting}$$(2)

Viscosity was estimated from the slopes of the linear parts of the graphs.

#### 2.2.2. Preparation of Feed Emulsions

Polysaccharide EO emulsion (PSC) and polymeric dispersion (PLM) both containing 30% *w*/*w* dispersed phase and 70% *w*/*w* water were prepared separately. The PSC emulsion contained 4% encapsulated EO in the dispersed phase, which consisted (*w*/*w*) of: AG 72%, MS 12%, and MD 12%. AG and MD were hydrated overnight at 5 °C, and MS was added at 82 °C followed by homogenization (circumferential velocity = 19.95 m/s, t = 20 min, Ultra-Turrax Model TP18/10S5, IKA, Staufen im Breisgau, Germany). The dispersion was cooled at 5 °C, and EO was added followed by a further 5 min of homogenization. The PLM dispersion contained 30% *w*/*w* Eudragit L100. Then, 30 g of polymer were dissolved in 70 g of medium consisting of 53.2 g of deionized water and 16.8 g of 1N NH_3_ (added to disperse the polymer). PLM was kept under magnetic stirring for 6 h and then added slowly under stirring to PSC emulsion to give a combined PLM/PCS emulsion. The concentration of dispersed phase in the feed PSC or PLM/PSC feed emulsions was always 30% *w*/*w* (28.8% encapsulating material plus 1.2% EO) selected on the basis of acceptable stability, viscosity, and minimal losses during spray drying.

The compositions of feed emulsions are given in Table 1. EO was microencapsulated either in polysaccharides only, i.e., PLM/PSC 0:100 (abbreviated PSC; emulsions: ADA_0_100_, SOS1_0_100_, SOS3_0_100_) or in PLM/PSC 80/20 (abbreviated PLM/PSC; emulsions: ADA_80_20_, SOS1_80_20_, SOS3_80_20_) aiming for enteric release. Higher PLM/PSC combinations although feasible were not considered, due to the low remaining EO content in the final spray-dried product (less than 4% *w*/*w*). Compositions for different PLM/PSC ratios can be calculated from the percentage of dispersed phase without EO (D′) (i.e., 30 − 1.2 = 28.8%) and the relative PLM content in the wall X (where X = 0 or 0.8) using Equations (3) and (4) for AG% and starch or MD% respectively.
AG% = (D′ − X)/1.33(3)
MS% or MD% = (D′ − X)/8(4)

The value 1.33 corresponds to the ratio of AG in the dispersed phase (=92/72) and 8 corresponds to the ratio of MS or MD in the dispersed phase (=92/12). EO% was always 1.2% in the feed emulsion or 4% *w*/*w* in the spray-dried final product.

#### 2.2.3. Droplet Size and Stability of Feed Emulsions

Droplet size distribution of the feed emulsions was analyzed with optical microscopy (Olympus BX41 microscope, Tokyo, Japan and video camera Leica DF295, Wetzlar, Germany) using image analysis software (Leica Microsystems, Heerbrugg, Switzerland). About 600 droplets were analyzed in 4–5 different fields at ×40 magnification. Droplet size was expressed as equivalent circle diameter (diameter of sphere having the same projected area as the measured droplets). The span of the size distribution was calculated from Equation (5) where d_10_, d_50_, and d_90_ were the diameters corresponding to 10%, 50%, and 90% of the distribution.
Span = (d_90_ − d_10_)/d_50_(5)

Stability was evaluated using an optical analyzer (Turbiscan^®^ MA Classic 2000, Formulaction, Toulouse, France) (850 nm light, scattered at 135°). The glass test tube was filled to 45 mm height and scanned at 40 μm intervals. Back scattering (BS%)—distance profiles were recorded automatically every 5 min for 45 min.

#### 2.2.4. Spray Drying

First, 100 mL of each feed emulsion was spray-dried with parallel air flow to the spray feed (B-191, Büchi, Flawil, Switzerland) under the conditions given in Table 2. Lower temperature was used for feed emulsions containing polymer (145 °C instead of 180 °C) to avoid Eudragit^®^ L100 degradation at 176 °C [23]. At the same time, the feed rate was reduced to increase drying efficiency. From the mass of powder collected at the receiving end, the vessel relative yield (%) was calculated from Equation (6).
(6)$$\mathrm{Yield}\;\left(\%\right)=100\times \frac{\mathrm{Weight}\;\mathrm{of}\;\mathrm{collected}\;\mathrm{powder}}{\mathrm{Weight}\;\mathrm{of}\;\mathrm{dispersed}\;\mathrm{phase}\;\mathrm{in}\;\mathrm{feed}\;\mathrm{emulsion}}$$

Additionally, oil retention (RT%) was determined as previously described [5] from the expression:(7)$$\mathrm{RT}\%=100\times \frac{\mathrm{EO}\;\mathrm{in}\;\mathrm{spray}-\mathrm{dried}\;\mathrm{product}}{\mathrm{EO}\;\mathrm{in}\;\mathrm{feed}\;\mathrm{emulsion}}.$$

#### 2.2.5. Particle Size, Shape, Morphology, Density, and Moisture Content

The particle size and shape of the spray-dried products were analyzed with optical microscopy as described above for emulsion droplet size. Small amounts of powders were dispersed in liquid paraffin, and about 500 particles were analyzed in 4–5 different fields at ×40 magnification. Particle size was expressed as the equivalent circle diameter. Particle shape was expressed as a roundness index using Equation (8) [24], which expresses both geometry and surface irregularity and has a minimum value of 1 for a sphere.
Roundness index = perimeter^2^/(12.56 × mean projection area)(8)

The particle surface morphology of the spray-dried powders was examined qualitatively with scanning electron microscopy (SEM) (20 kV beam electron voltage, JSM 840A, Jeol, Tokyo, Japan).

Particle density was measured using helium pycnometry (Ultrapycnometer 1000, Quantachrome Instruments, Boynton Beach, FL, USA). The instrument was calibrated with a standard 7.0699 cm^3^ steel ball. Samples were accurately weighed (3 decimals) and purged for 3 min before measurement. Sample volume (average of 10 runs) was measured from the displaced gas. Measurements were made in triplicate.

The moisture content (MC% dry basis) of unprocessed materials and spray-dried powders was determined by thermogravimetry (TGA-50 analyzer, Shimadzu Corporation, Kyoto, Japan). Accurately weighted samples of about 5 mg were heated in the range of 30–180 °C at 10 °C/min under 50 mL/min nitrogen purge gas flow. MC% was calculated from the weight loss at 105 °C where the curves flattened.

#### 2.2.6. Electrostatic Charge of Spray Dried Powders

The electrostatic surface charges of spray-dried powders were measured with a resolution of 1 pC using a Faraday pail (JCI 140, Chilworth Technology Ltd., Southampton, UK) connected to a static monitor (JCI 147, Chilworth Technology Ltd., Southampton, UK). For the measurement, about 1 g of powder previously kept in closed glass bottles was dropped into the center of the pail cylinder. The charge was read directly on the instrument monitor as nC.

#### 2.2.7. Dynamic Packing and Angle of Repose of Spray-Dried Powders

Packing densities (bulk and tap after 300 taps) were determined using a tap tester (14 mm vertical drop, Erweka SVM 101, USP1, Heusenstamm, Germany) fitted with a 25 mL glass cylinder. From the bulk (ρ_b_) and tap (ρ_t_) densities, the Carr’s index (CC%) and Hausner ratio (HR) were calculated [25,26].
CC% = 100 × (ρ_t_ − ρ_b_)/(ρ_t_)(9)
HR = ρ_t_/ρ_b_(10)

The limitation of these two indices is that they represent only two states of the powder bed: initial and final. Therefore, their discriminating ability for the present experimental powders with similar particle size is inadequate. For this reason, the dynamic packing behavior of the powders was evaluated by recording volumetric changes every 10 taps up to 300 and processing the results using different packing models. These models have been found to provide useful information on the behavior of fine pharmaceutical powders for powder materials in similar particle size range with the present spray-dried powders [27].

Kawakita’s model [28] has been extensively used and is described with Equation (11).
N/c = N/a + 1/a × b(11)

N is the number of taps, and c expresses the maximum relative volume reduction given by Equation (12). The constant a is related to packing ability, and constant b is related to powder cohesiveness. Vo is the initial volume and V_300_ the volume after 300 taps.
c = (Vo − V_300_)/Vo(12)

By plotting N/c versus N constants, a and b can be derived using linear regression analysis.

The Varthalis–Pilpel [29] model is able to differentiate the behavior of pharmaceutical powders during tapping and is expressed by Equation (13)
ε^2^/(1 − ε) = A_IF_ × N + Κ_O_(13)
where the left side of the equation is denoted for convenience K [=ε^2^/(1 − ε)]. ε is the powder bed porosity expressed by Equation (14). By plotting K versus N, the slope of the straight line is obtained as A_IF_ and Ko is obtained as the intercept on the ordinate. A_IF_ is termed as the angle of internal flow and is related to the ability of particles to flow past each other during tapping and hence to their packing ability. As A_IF_ increases, packing ability decreases.
ε = 1 − (ρ_N_/ρ_s_)(14)

Mohammadi-Harnby [30] has been found to be related to the microstructure of the powder bed. It is expressed by Equation (15)
ρ_N_ = ρ_t_ − D × exp^(−N/T)^(15)
where ρ_N_ is the powder bed density after N taps, ρ_s_ is the particle (or ‘true’) density, and D equals (ρ_t_ − ρ_b_). T is a constant derived by fitting the data to the above exponential equation, and its value reflects the microstructure of the powder bed. Low T values indicate the presence of independently behaving particle units, whereas high values agglomerated the powder state.

The angle of repose provides an estimation of the flowability of powders at an intermediate state between cohesive and free flowing [31]. For the measurement, powder was placed inside an open-ended metallic cylinder (5 cm × 0.5 cm), which was subsequently raised carefully at a rate of about 24 cm/min, allowing the stacked powder to flow out from the sides to form a cone-like heap on the rubber base. Cone dimensions were obtained either from the physical dimensions using calipers or by image analysis of the cone projected area using images taken at a fixed 15 cm distance. The angle of repose (*θ*) was obtained from Equation (16), where r is the radius of the cone base and h is the hypotenuse of the right triangle formed with the height of the cone as the opposite and its radius as the adjacent leg.
cos*θ* = r/h(16)

#### 2.2.8. FTIR Spectroscopy

FTIR spectra were obtained using a Shimadzu FT-IR-Prestige-21 spectrometer (Shimadzu Corporation, Tokyo, Japan) supported by software (Shimatzu IR solution 1.3). The instrument was attached to a horizontal Golden Gate MKII single-reflection ATR system (Specac, Kent, UK) equipped with a Diamond/ZnSe crystal (45° angle to the beam, 1.66 μm at 1000 cm^−1^ depth of penetration, 2.4 refractive index, and 525 cm^−1^ long wavelength cut-off). Unprocessed materials or spray-dried powder were placed on the diamond disk using a sapphire anvil to restrain the powder. The analyzed spectra were an average of 64 scans collected at 4 cm^−1^ resolution.

#### 2.2.9. Reconstitution of Emulsions from the Spray-Dried Powders

The ability of spray-dried powder to re-emulsify in liquid is a prerequisite for exerting its antimicrobial activity. Powder samples corresponding to 2.8 mg EO were placed in 200 mL of distilled water in a USP II apparatus vessel (Pharma Test PTW 2, Hainburg, Germany) at 37 ± 0.5 °C and 100 rpm agitation rate. Aliquots (3 mL) were withdrawn at timely intervals from 5 to 240 min and analyzed for transmittance (T%, 850 nm) using the optical analyzer (Turbiscan^®^ MA Classic 2000, Formulaction, Toulouse, France) described above for stability testing but operated in transmittance mode [5,32]. The instrument was calibrated for T% = 100 with special silicon oil. After measurement, the samples were returned to the vessel.

#### 2.2.10. In-Vitro Release of EO from the Spray-Dried Powders

Samples of spray-dried powders corresponding to 10 mg of EO (nominal amount) were sealed in dialysis cellulose membranes (cut-off 12500, Sigma-Aldrich, St. Louis, MO, USA) and immersed in the USP Apparatus II vessel (Pharma Test PTW 2) containing 200 mL of pH 1.2 buffer solution (100 mL 0.2 N KCl, 170 mL 0.2 N HCl, and 130 mL deionized water), at 37 ± 0.5 °C and 100 rpm agitation rate [5]. After 120 min of testing, the medium pH was changed to 6.8 by adding 8.06 g of sodium phosphate (NaH_2_PO_4_). Aliquots were taken at timely intervals from 5 to 240 min and analyzed for EO by UV spectroscopy (Pharma Spec UV—1700, Shimadzu, Kyoto, Japan) at 273 nm. Tests were repeated in triplicate.

#### 2.2.11. Experimental Design and Statistical Analysis

A 2 × 3 factorial design was used to examine the effects of microencapsulating wall consistency PLM/PCS 0/100 or 80/20 (enteric) and starch grade in the PSC on the technological properties of spray-dried products. The results were analyzed using SPSS 20 software (IBM Inc., Chicago, IL, USA, Version 25.0, 2019).

## 3. Results and Discussion

### 3.1. Surface Activity and Viscosity of Modified Starches

The quality of spray-dried EO products depends on the properties of feed emulsions. The surface activity of the wall ingredients is expected to affect their arrangement at the oil/water interface. In the microencapsulating wall, only the modified starches have surface activity and to a small extend Arabic gum due to the small glycoprotein content. From the structures of the two chemically different modified starches shown in Figure 1, it appears that sodium octenyl succinate (SOS) has an amphiphilic structure consisting of a hydrophilic part of glucopyranose units and succinate ester, and a hydrophobic part of octenyl tail, therefore possessing surface activity. Its HLB value was calculated from Equation (17) [33] using as the degree of substitution DS = 2.05% (Roquette, personal communication) and was found to be 12.66.
HLB = 12.7 − 1.49 DS(17)

On the contrary, acetylated di-starch adipate (ADA) does not have a hydrophobic part, consisting of hydrophilic glucopyranose units, acetyl, and adipate groups, and hence lacks surface activity. This is demonstrated in Figure 2, showing plots of water/essential oil interfacial tension versus starch concentration. The two SOS starch grades decreased greatly the oil/water interfacial tension with concentration, whereas ADA did not [34]. At very low concentrations, the decrease of SOS3 is greater than SOS1. This is attributed to the faster diffusion of the low viscosity SOS3 grade to the interface, leading to a rapid lowering of tension compared to the high viscosity SOS1. At higher concentrations, the two grades reached similar interfacial tension. Therefore, the three starch grades are expected to influence the formation of the encapsulating wall differently.

In Figure 3, shear stress versus shear rate plots are shown for aqueous solutions of the modified starches covering concentrations from 3.6% *w*/*w* at which starch is present in the feed emulsions up to 10% *w*/*w*. In all cases, above shear rate 30 s^−1^, there is a linear increase of shear stress with shear rate, indicating Newtonian behavior. The slope expressing viscosity increases with the concentration of the dispersed phase. For the tested concentrations (3.6%, 7.4%, and 10% *w*/*w*), ADA solutions gave the lowest slopes or viscosities (1.67, 1.95, and 2.53 cP), followed by SOS3 (2.28, 3.93, and 5.10 cP) and SOS1 (3.25, 4.93, and 7.56 cP). Since starch is only present at a small percentage in the feed emulsions (3.6% in the PCS and 0.6% in the PSC/PLM), the differences in viscosity have a minor impact on droplet formation. However, it may be assumed that these differences may affect the molecular arrangement in the encapsulating wall due to the different molecular mobility. This will assist the intermolecular packing of the PSC with PLM.

### 3.2. Characterization on Feed Emulsions

In Figure 4, images of the six feed emulsions of the three starch grades stabilized with PSC (emulsions ADA_0_100_, SOS1_0_100_, SOS3_0_100_), or with PLM/PSC (ADA_80_20_, SOS1_80_20_, SOS3_80_20_) are shown together with corresponding median droplet diameter and distribution span values. Comparing the two chemically different starches, it appears that SOS provided more uniform PSC emulsions (lower span) and smaller d_50_ than ADA. However, these differences were diminished when the combination of PLM/PSC was used for encapsulation. PLM/PSC emulsions had slightly smaller d_50_ (9.9–10.8 μm compared to 11.5–12.3 μm) and lower span (1.6–1.8 compared with 1.9–2.3), indicating more uniform size distribution and emulsion stability.

In Figure 5, the results of turbidimetry for the six experimental feed emulsions are shown as back scattering (%) versus distance profiles, which were recorded every 5 min over a 45-min period (total dispersed phase 30% *w*/*w*). In all cases, the shape and position of the profiles remains unchanged over time, indicating good stability. PLM/PCS emulsions showed a greater scattering of about 90% compared to 30–40% of the PSC, which may be partly ascribed to the smaller particle size (Figure 4) and partly to the greater light diffusivity of the brighter, white color PLM/PCS emulsions, compared to the pale-yellow color of PSC [35].

### 3.3. Spray-Dried Products

#### 3.3.1. Production Yield, Moisture Content Oil Retention, and Particle Density

In Table 3, the results of production yield, EO retention, moisture content (MC%), and physical properties (particle density, size distribution, shape, and particle surface charges) of the spray-dried products prepared with PSC feed emulsion (SD-ADA_0_100_, SD-SOS1_0_100_, SD-SOS3_0_100_) or PLM/PSC emulsion (SD-ADA_80_20_, SD-SOS1_80_20_, SD-SOS3_80_20_) are presented. Complimentary, SEM images are shown in Figure 6. Spray drying on lab-scale equipment yields lower recovery and smaller particle size compared to large scale, so the results should be interpreted in a comparative perspective.

From Table 3, it can be seen that for the PSC products, the yield was between 25% and 40.8%, whereas for the PLM/PSC, it was higher: between 29.9% and 49.9%. On the other hand, from Table 3, it is seen that the MC% of the PSC products was more than double that of PLM/PCS (12.0–12.4% compared with 5.1–6.0%). This is ascribed to the hydrophilicity of polysaccharides reflected on their high MC% values which are 10.0% for AG, 4.9% for MD, and 9.0%, 6.2%, and 6.1% for ADA, SOS1, and SOS3 starches, respectively. Among PSC or PLM/PSC products, MC% was not significantly different. The lower yield of the PSC products can be attributed to the softer polysaccharide wall and to their high MC%, since both of them promote adhesion to the drying chamber walls, resulting in a loss of material [36,37]. Considering the effects of starch grade, higher yield in both the PCS and PLM/PCS products was obtained from SOS1 (40.8% and 49.9%), followed by SOS3 (33.5% and 39.6%) and ADA (25.0% and 29.9%). The higher yield of SOS1 should be associated with its higher viscosity or higher molecular weight (Figure 2 and Figure 3). Since the moisture contents of the products in the PCS or PLM/PCS groups were similar, higher viscosity signifies a greater glass transition temperature and less sticking behavior of SOS1 products to the drying chamber wall [38].

From Table 3, it is seen that the oil retention was between 67.5% and 72.2% for the PSC products and slightly higher for PLM/PSC: between 72.8% and 73.5%. This is attributed to the lower temperature applied during the spray drying of PLM/PSC products (145 °C compared with 180 °C). The particle densities ranged from 1.249 to 1.369 g/cc for the PSC and 1.383 to 1.414 g/cc for the PLM/PSC products. The greater particle densities of the latter indicate less ‘blind’ void spaces in the particles or easier vapor escape from the particle surface during spray drying [39].

#### 3.3.2. Particle Morphology and Size Distribution

In Figure 6, SEM images of spray-dried powders of the three starch types prepared from emulsions stabilized with PSC only (SD-ADA_0_100_, SD-SOS1_0_100_, SD-SOS3_0_100_) or PLM/PSC (SD-ADA_80_20_, SD-SOS1_80_20_, SD-SOS3_80_20_) are shown. They consist of mixtures of single particles and particle agglomerates. Powders prepared with PSC contain a large number of shriveled particles and agglomerates, which is reflected on the higher shape index values between 1.96 and 2.17 compared with 1.64 to 1.81 for the PLM/PSC (Table 3). In particular, the presence of polymers in the encapsulating wall (products SD-ADA_80_20_, SD-SOS1_80_20_, SD-SOS3_80_20_) resulted in the smoothing of particle surfaces by eliminating the cavities seen in the PCS products.

This is related to two processes. One is the rate of liquid evaporation from the drying droplet relative to the diffusion rate of dispersed phase to the center of the droplet, which is expressed by the dimensionless Peclet number (Pe) (Equation (18)) [40]. A high Pe number signifies the accumulation of dispersed phase at the surface and shrinkage.
(18)$$\mathrm{Pe}=\frac{{\left(2R\right)}^{2}}{\tau_{d}D_{f}}$$

R is the droplet radius, D_f_ is the diffusion rate, and τ_d_ is the drying time of the droplet. In the present study, Pe and particle formation should be mainly influenced by the drying time, since the higher drying temperature applied for PSC emulsions (180 °C, Table 2) compared to PSC/PLM (145 °C, Table 2) leads to a greater evaporation rate. Therefore, for PSC emulsions, drying is faster, τ_d_ is lower, and Pe is greater compared to PSC/PLM. The other factor influencing particle formation is the consistency of the encapsulating wall. It is known that the viscoelastic behavior of polysaccharides leads to the formation of a flexible surface skin in the late stages of drying, causing localized collapse and surface cavities [5,41]. The combination of Eudragit^®^ L100 with PSC results in the formation of a more rigid encapsulating wall, thus avoiding cavitation [42].

Evidence of the physical interaction of the polymer (PLM) with the polysaccharides (PSC) and its physical influence on the characteristics of the encapsulating wall is provided from the values of electrostatic charges presented in Table 3. Powders prepared with PCS only are negatively charged (from −0.89 to −0.64 nC), but in the presence of the polymer, the charges decrease (in absolute values) to near zero (−0.02 to −0.14 nC). The negative charge on the PSC powers is attributed to the presence of negatively charged carboxyl groups of amino acids and glucuronic acid residues on aminogalactane blocks of the Arabic gum [5,43]. Similarly, Eudragit^®^ L100 being an anionic polymer with carboxyl groups would be expected to be negatively charged. Therefore, the reduction of electrostatic charges of the PLM/PSC powders may be ascribed to a reorientation of the carboxyl groups on the particle surface. The standard deviations varied from 0.08 to 0.11 for the PSC and from 0.08 to 0.11 for the PSC/PLM product.

From the particle size distribution results in Table 3, it appears that the median particle diameters of PSC and PLM/PSC products were similar (d_50_ from 9.2 to 10.7 μm and 9.3 to 11.6 μm, respectively). This was expected, since the only difference in the preparation of PSC and PLM/PSC batches was the drying temperature. However, the distributions of the PLM/PSC powders were narrower (span between 0.51 and 0.91 compared to 0.70 to 0.99). These results follow the same trend of the feed emulsions droplet size distribution (Figure 4).

#### 3.3.3. Powder Packing Evaluation

In Table 4, bulk (ρ_b_) and tap (ρ_t_) densities are presented together with the results of the packing properties of the experimental spray-dried essential oil powder products. It can be seen that those prepared from PLM/PSC feed emulsions have higher ρ_b_ and ρ_t_ values (0.38 to 0.40 g/cc compared with 0.27 to 0.28 g/cc and 0.62 to 0.64 compared with 0.51 to 0.56 g/cc, respectively), implying that in the final dosage form, more powder weight can be filled per unit volume compared to PSC. The degree of consolidation or packing is expected to be affected by the electrostatic charges of the powders. In general, PLM/PSC powders with nearer to zero charges (from −0.02 to −0.14 nC, Table 3) are expected to pack more densely than PSC powders carrying more charge (from −0.64 to −0.89 nC) due to the repulsive forces of the latter. This is clearly seen in Table 4 from the higher ρ_b_ and ρ_t_ values of PLM/PSC powders.

The Hausner ratio and Carr’s index are lower for the PLM/PSC (1.61 to 1.65 compared with 1.92 to 2.00 and 37.9% to 39.5% compared with 47.9% to 50.0%, respectively), indicating greater packing ability. However, although the last two indices are easily determined, they describe only two states of the powder bed at the beginning and at the end of the packing process. As a result of this, when powders of similar particle size are compared, they may not be sensitive enough to detect the differences [44]. For example, in Table 4, the differences in Hausner or CC% values among products of different starch grade that belong to the PSC (SD-ADA_0_100_, SD-SOS1_0_100_, SD-SOS30_100) or to the PLM/PSC group (SD-ADA_80_20_, SD-SOS1_80_20_, SD-SOS3_80_20_) are within the experimental error. Therefore, to elucidate any differences between them, the dynamic models described by Equations (11), (13), and (15) which are derived from a greater number of measurements were applied.

#### 3.3.4. Dynamic Packing Models

As a result of particle morphological differences, the packing behavior and flowability of the spray-dried powder products are expected to differ. They are fine powders (d_50_ of about 10 μm, Table 3) but with more or less round-shaped particles (Figure 6). Therefore, their behavior is intermediate between cohesive and free flowing [31,45,46], and even small morphological differences can produce incremental changes, which may prove important for packing and flowability [28]. Therefore, the discriminating ability of the applied models is of particular interest.

In Figure 7A, plots representing the three dynamic packing models are presented. In the Kawakita plots, the points fall in straight lines, confirming model linearity. The lines form two groups. One group consisted of EO powders prepared with PSC only (SD-ADA_0_100_, SD-SOS1_0_100_, and SD-SOS3_0_100_) with parameter ‘a’ values between 0.54 and 0.57, and a second group consisted of PLM/PSC batches (SD-ADA_80_20_, SD-SOS1_80_20_, and SD-SOS3_80_20_), with lower ‘a’ values between 0.46 and 0.48 indicating better packing ability.

In the Varthalis–Pilpel plots, the points of each product appear to fall in two continuous straight lines, implying two consolidation mechanisms: an initial representing the collapsing of weak irregular structures such as bridges and arches resulting in the fast filling of voids and consolidation, and a second mechanism promoting slower consolidation by particle rearrangement. PSC powders developed the break after 27 to 36 taps, whereas PLM/PSC powders developed the break earlier after 21 to 26 taps, acquiring uniformity of the powder bed faster. Since the investigated powders had particles in similar size ranges, this break point should be associated with differences of particle morphology and the degree of agglomeration of the spray-dried products. From the values of internal flow (AIF, Table 4), it can be seen that the PLM/PSC powders had lower values than PSC (21.1° to 29.0° compared with 31.3° to 42.3°), implying less frictional forces and easier packing for the PLM/PSC powders [29,45].

In the Mohammadi–Harnby plots, the points follow an exponential decrease, with the model lines providing good fitting. Values of the parameter T (Equation (7)) are presented in Table 4, and it can be seen that overall, they are lower for the PLM/PSC powders (from 74.7 to 106.8 compared with 109.4 to 114.2). T is related to the microstructure of the powder bed. Lower values reflect easier packing of a structure consisting of rigid single particles or rigid particle agglomerates, i.e., PLM/PSC particles due to the presence of Eudragit L100 in the encapsulating wall, whereas higher T values reflect weakly agglomerated particles, i.e., PSC particles with packing difficulty.

The above results are in agreement with the extensive degree of agglomeration particles in the PSC powders (SEM, Figure 6) and clearly demonstrate the technological superiority of the PLM/PSC over PSC products in terms of packing indices.

Furthermore, considering the values of the model parameters Kawakita a, AIF, and T in Table 4, it seems that for the PLM/PSC EO powders, AIF and T follow the same decreasing order: SD-ADA_80_20_ > SD-SOS1_80_20_ > SD-SOS3_80_20_. To elucidate and compare the effects of the type of microencapsulating wall (PSC or PLM/PSC) and type of starch (ADA, SOS1, SOS3) on AIF and T, statistical analysis was performed using a factorial experimental design of two factors: type of microencapsulating wall (two levels: PSC, PLM/PSC) and type of starch (three levels: ADA, SOS1, and SOS3) (General Linear Model, SPSS 20.0). Further information of the statistical analysis can be found elsewhere [47]. Significant interactions were found for the effects of the two factors on both dynamic packing parameters and also on particle shape and tap density. Interaction plots are depicted in Figure 8, where it can be seen that it is the SD-SOS3_80_20_ batch (green line) that exhibits the largest drop in the values of the measured properties compared to the other starch batches ADA and SOS1, causing the statistical interaction (for particle shape significance, *p* = 0.015; for tap density, *p* = 0.003; for AIF, *p* = 0.001; and for constant T, *p* = 0.011). This means that the incorporation of starch SD-SOS3_80_20_ makes the presence of Eudragit L100 more effective in altering the packing properties of the spray-dried products. From the above results, it becomes clear that the presence of the polymer in the encapsulating wall, and especially together with the surface active low viscosity starch grade SD-SOS3 results in great improvement of the packing ability of the spray-dried powders, which is very important for further processing.

#### 3.3.5. Angle of Repose

The angle of repose is a property related to the resistance to movement between particles and is described in the 30(6) Harmonization: <1174> POWDER FLOW document of the USP [48]. Despite some experimental uncertainties, it is extensively used in the pharmaceutical industry as an index of flowability, as it is easy to perform and can predict manufacturing problems at an early stage. Representative images of experimental conical heaps of EO spray-dried powders encapsulated in the PLM/PSC wall are presented in Figure 7B and *θ*° values are given in Table 5. They were calculated either from direct measurements of the cone physical dimensions or from the mean projected area by image analysis. PSC powders did not form conical heaps, and hence the angle of repose could not be estimated. When the formed cone was incomplete, the apex was obtained by extrapolation of the sides. PCS spray-dried powders could not form conical heaps at all, and the results are not presented in Figure 7B and Table 5. This emphasizes again the positive influence of Eudragit^®^ L100 in the quality of encapsulating wall by enabling the formation of particles with smoother surface and neutral electrostatic charges (Table 3), resulting in improved flow evidenced by a measurable angle of repose.

Furthermore, from Figure 7B, it can be seen that among the PLM/PSC powders, SD-ADA_80_20_ containing the non-surface active grade ADA formed a conical heap with a partly collapsed side and without the top, SD-SOS1_80_20_ containing the surface active/high viscosity SOS1 grade formed a cone without the top, and only the surface active/low viscosity grade SOS3 formed a complete cone. The *θ°* value of the latter is 42.5° (Table 5) which is near 40°, and this is the suggested threshold for a free-flowing powder [31]. From the above results, it can be assumed that grade SOS3 assists the combination of the polymer and PSC phases in the formation of a rigid encapsulating wall, thus minimizing particle aggregation and improving particle sphericity and flowability (Figure 6). From Table 5, it appears that image analysis gave higher *θ* values, although these were of the same ranking as the physical measurement.

### 3.4. Fourier-Transform Infrared Spectroscopy (FTIR)

To elucidate whether besides the physical PLM and PSC interaction evidenced by electrostatic charge neutralization there is chemical interaction, FTIR spectra of the spray-dried products with or without polymers were compared. In Figure 9, the spectra of wall constituents and spray-dried products prepared at PLM/PSC ratios 0/100 and 80/20 with starch grades ADA and SOS3 are presented (SOS1 spectra were not included because they were the same with SOS3).

The Eudragit^®^ spectrum shows a broad band between 2800 and 3050 cm^−1^ due to methylene CH stretch, a peak near 1710 cm^−1^ due to acrylic ketone (C=O stretch), at 1249 and 1153 cm^−1^ due to ketone acyclic (C=O stretch), at 1061 cm^−1^ due to ester C-O and smaller peaks at 1390 and 1482 cm^−1^ due to C-H deformation [49]. The spectra of polysaccharides show a broad band in the region of 3050–3600 cm^−1^ due to a hydrogen-bonded OH group, a band in the region 2800–3000 cm^−1^ due to a carboxylic group, at 1600 cm^−1^ (AG) or 1650 cm^−1^ (starches) due to C=O stretching, at 1400 cm^−1^ due to CH_3_ and CH bending, and at 1052 cm^−1^ (AG) or 1000 cm^−1^ (starches, MD) due to OH deformation [50].

To examine possible interactions between the PLM and PSC constituents, FTIR spectra of products SD-ADA_0_100_ and SD-SOS3_0_100_ prepared with polysaccharides (PSC) emulsion are compared in Figure 9b with the corresponding spectra of SD-SOS_80_20_ and SD-ADA_80_20_ prepared with PLM/PSC 80/20 emulsion. The presence of PLM is pronounced in the PLM/PSC products at 1710 cm^−1^. The presence of PSC is less pronounced in these products due to its lower content (for the PLM/PSC ratio 80/20: Arabic gum is 12%, starch and maltodextrin 2% *w*/*w*, Table 1), but it is clearly evidenced from the broad band at 3050–3600 cm^−1^ and the shoulder kink at 1600 cm^−1^, which is shifted to a slightly higher wavenumber 1610 cm^−1^, implying weak H-bonding interaction of the C=0 with the carboxyl OH of the polymer. A smaller shoulder kink attributable to the left part of the broad PSC peak at 1240 cm^−1^ is also present in the SD-ADA_80_20_ and SD-SOS_80_20_ spectra (more discernible in the former). From these results, it is evident that the peak positions of the PSC constituents and polymer remain virtually the same in the spectra of the PLM/PSC products, and there is no significant interaction between them. Consequently, the co-existence of PLM with PSC is not expected to affect the gastro-protective action of Eudragit L100 during dissolution of the spray-dried powders prepared with the PLM/PSC encapsulating combination, which is an important goal of the study.

### 3.5. Reconstitution

Reconstitution of a dry emulsion upon contact with liquid into its original oil/water emulsion is an important step before releasing the encapsulated ingredient [5,32]. This is because the formation of numerous droplets provides a very large surface in contact with the liquid so that if the encapsulating wall does not pose a barrier to the penetration of water, release takes place. In Figure 10, plots of transmittance of reconstitution medium versus time are presented for the spray-dried products. A rapid decrease of the percentage of transmittance is seen in the pH 1.2 medium for the PSC products during the first 60 min, indicating re-emulsification. Although the drop is similar for the products of the three starch grades, the final transmittance is greater for ADA (53%) followed by SOS1 (47%) and SOS (34%). This sequence should reflect the number of finally formed droplets. The surface-active grades SOS are able to reduce the oil/water interfacial tension and produce finer dispersions or a greater number of droplets. Alternatively, these differences may have been caused by the differences in oil retention, although these were rather small. Between the two SOS grades, the lower viscosity SOS3 grade shows a markedly lower final percentage of transmittance. On the other hand, PLM/PSC products show a negligible transmittance change at pH 1.2. This is explained by the domination of the encapsulating wall by the polymer molecules present at a greater ratio of 80/20, which inhibits the contact of the medium with the PSC components. However, when the pH of the medium was changed to 6.8%, transmittance dropped in a similar manner with the drop in the corresponding PSC products at pH 1.2, indicating access of the liquid to PSC and reconstitution. Again, the finally achieved transmittance followed the same sequence of the corresponding PSC products, i.e., SD-ADA_80-20_ > SD-SOS1_80-20_ > SD-SOS3_80_20_.

### 3.6. In Vitro Release of EO from the Spray-Dried Products

In Figure 11, the release profiles of EO from the six experimental spray-dried products are presented. The dialysis membrane method was used for testing, for the first 120 min at pH 1.2, after which it changed to 6.8 by the addition of sodium phosphate. This method is useful for testing systems where the release of colloidal particles interferes with EO analysis. The method has been criticized for leading to the misinterpretation of release kinetics mainly of lipophilic drugs due to reversible binding [51,52]. However, in the present work, the passage of EO across the dialysis membrane into the dissolution medium is not expected to be significantly affected by the presence of the membrane because of the hydrophilic nature and low molecular weight of the EO ingredients (i.e., Carvacrol MW = 155.22, p-Cymene MW = 139.21, γ-Terpinene MW = 136.23, and Thymol MW = 150.22).

In Figure 11, the release profiles of EO are presented. Considering that a nominal 10 mg of EO was added in the tested samples and about 70% of the EO remained after spray drying (Table 3), the PSC products resulted in complete EO release (approximately 7 mg) in gastric medium within 120 min. This is in agreement with previous results [5] confirming the reproducibility of the applied dialysis method. On the other hand, spray-dried products prepared with PLM/PSC provided good gastric protection by releasing only a small EO amount of about 15 % (relative to 7 mg) over a period of 120 min. The EO released from the PLM/PSC products at pH 1.2 approaches the content of PSC in (PLM/PSC 80/20) the encapsulating wall. Furthermore, from Figure 11, it can be seen that in the pH 6.8 medium, all the tested batches released EO completely. The effective gastric protection of the PLM/PSC products is attributed to two reasons: (1) to the physical polymer/polysaccharides interactions promoting close molecular packing, as indicated by the neutralization of electrostatic surface charges and by the smoothing of the particle surfaces, and (2) to the absence of significant chemical interactions allowing the polymer carboxyl groups to function by keeping the wall impermeable to water at pH 1.2, but allowing penetration due to ionization and repulsion of the carboxyl groups at pH 6.9.

## 4. Conclusions

A promising enteric release formulation of oregano essential oil (EO) was developed by following a two-step robust process: (1) preparation of feed emulsions stabilized by a combined Eudragit L100/polysaccharide (PLM/PSC) encapsulating wall and (2) spray drying of the emulsions to give powders with encapsulated oil. This combined process utilizes the ability of polysaccharides to stabilize emulsions, which is well known in the food industry with the ability of methacrylic polymers to prevent release in gastric fluids and deliver the active ingredient in the intestine, which is extensively practiced in the pharmaceutical industry. Powder products with good flowability were obtained when the surface-active, low-viscosity sodium octenyl succinate starch was included in the PSC composition. Physical interaction between PLM and PSC was demonstrated by the neutralization of electrostatic charges by the PLM and by the absence of chemical interactions, as evidenced by the spectral analysis. Re-emulsification from the PLM/PSC spray-dried powders at pH 1.2 was negligible, and EO release was only about 15%, confirming gastric protection. This is due to the availability of sufficient free functional carboxyl groups of the polymer. At pH 6.8, re-emulsification and EO release were completed within one hour, confirming the ability of PSC to re-emulsify and release EO. As far as we know, this is the first attempt to produce enteric oral forms of an essential oil. The solid form enables the efficient administration of an accurate dose to the intestine where pathogens mostly reside.

## Figures and Tables

**Figure 1 pharmaceutics-12-00571-f001:**
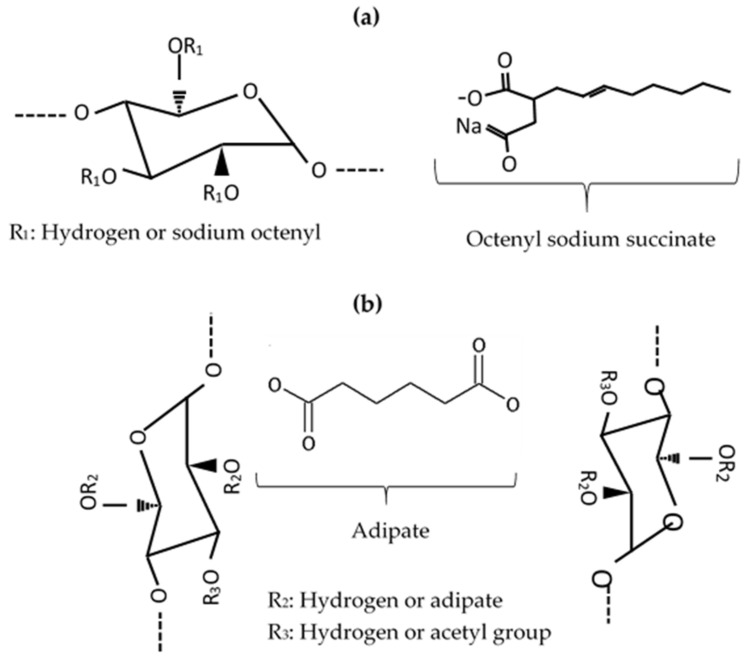
Chemical structures of starches: (**a**) Sodium octenyl succinate (SOS) (C_6_H_10_O_5_)x(C_12_H_18_O_3_Na)y and (**b**) Acetylated distarch adipate Starch-O-R-O-Starch (ADA), where R_2_ = CO-(CH_2_)_4_-CO and R_3_ = –COCH_3_ (modified from [22]).

**Figure 2 pharmaceutics-12-00571-f002:**
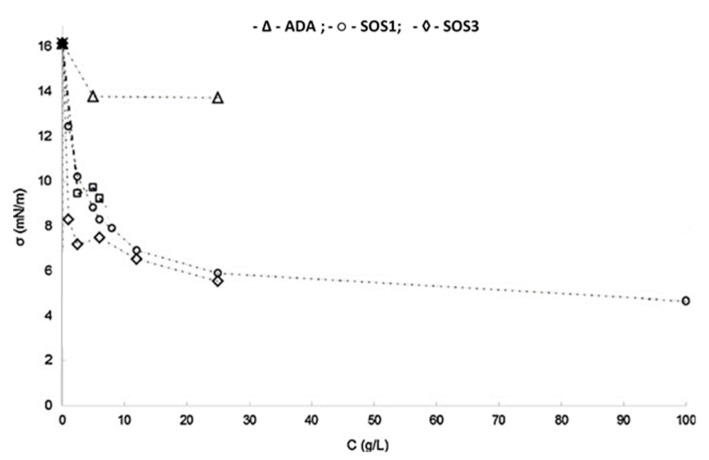
Reduction in interfacial tension between water and rosemary essential oil with the addition of ADA (-Δ-), SOS1 (-o-), and SOS3 (-◊-) (adapted with permission from [34]).

**Figure 3 pharmaceutics-12-00571-f003:**
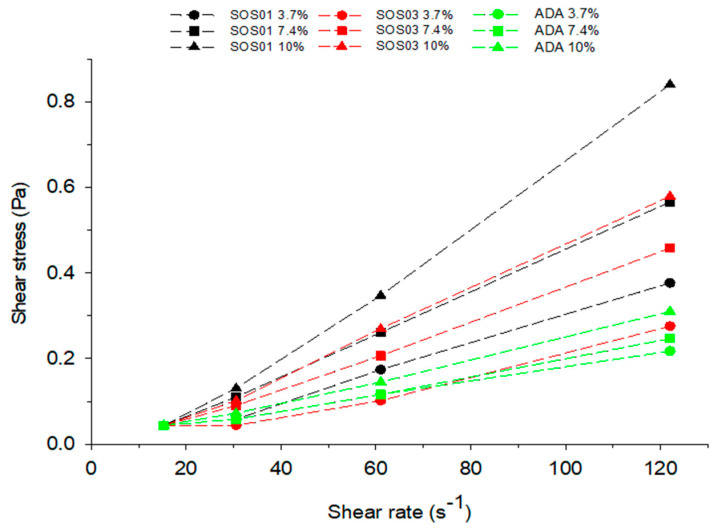
Shear stress vs. shear rate plots for starch grades of sodium octenyl succinate starches (SOS: high viscosity, SOS1 and low viscosity, SOS3) and acetylate di-starch adipate (ADA).

**Figure 4 pharmaceutics-12-00571-f004:**
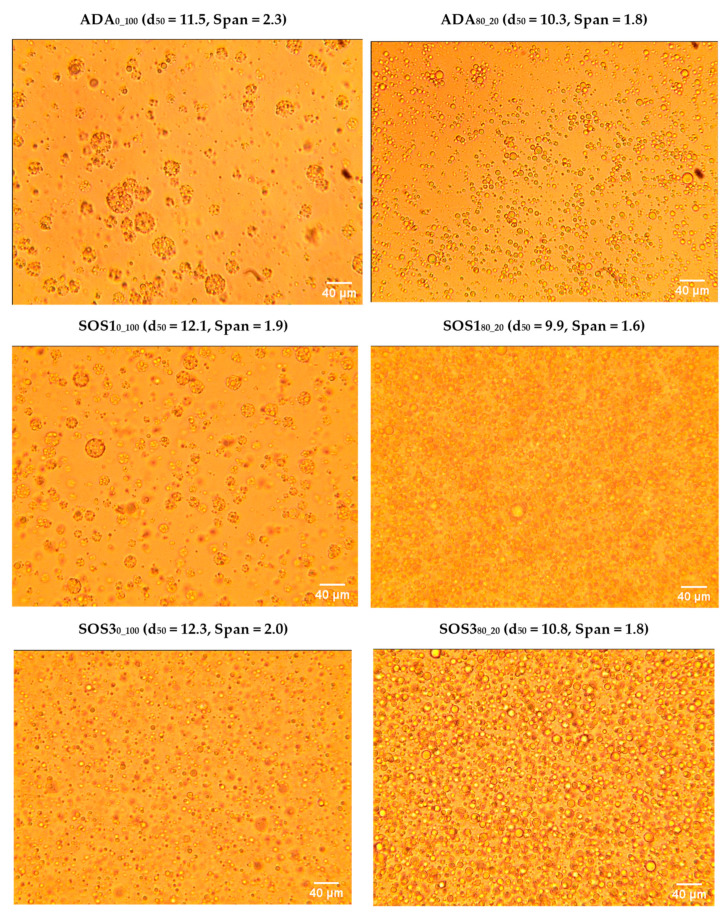
Optical microscopy images with a scale bar of feed emulsions of the three starch grades ADA, SOS1, and SOS3 stabilized with polysaccharides only (ADA_0_100_, SOS1_0_100_, SOS3_0_100_) or polymer/polysaccharides 80/20 (ADA_80_20_, SOS1_80_20_, SOS3_80_20_).

**Figure 5 pharmaceutics-12-00571-f005:**
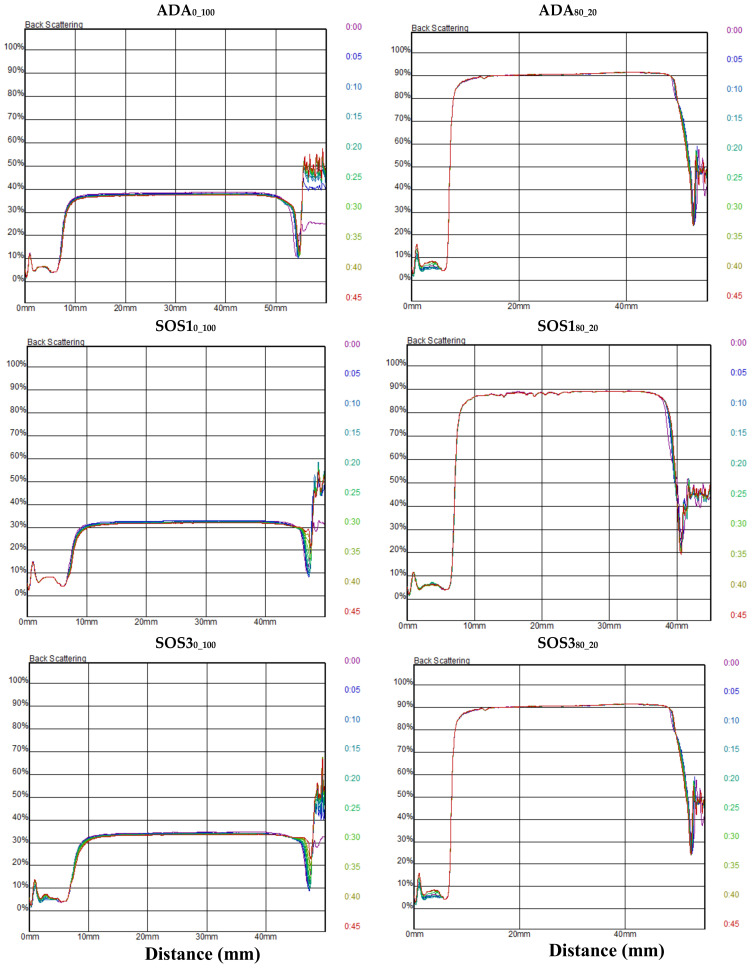
Back scattering (*Y*-axis, %) vs. distance (*X*-axis, mm) profiles of feed emulsions of the three starch grades: ADA, SOS1, and SOS3 stabilized with polysaccharides only (ADA_0_100_, SOS1_0_100_, SOS3_0_100_), or polymer/polysaccharides 80/20 (ADA_80_20_, SOS1_80_20_, SOS3_80_20_). (Different color curves correspond to different measurements taken at increasing 5 min time steps).

**Figure 6 pharmaceutics-12-00571-f006:**
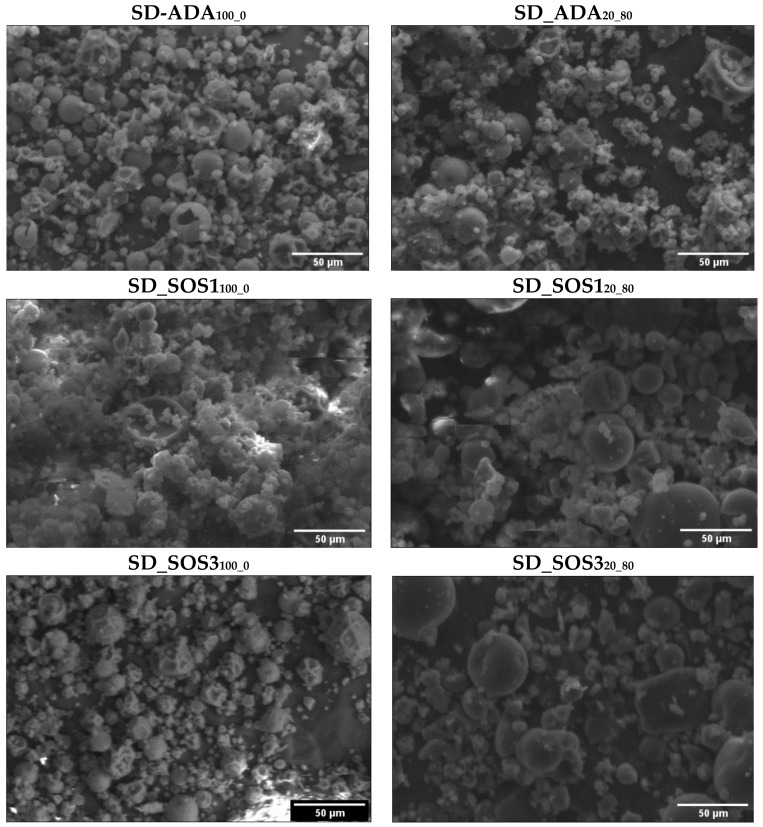
Scanning electron microscopy images of spray-dried EO products of the three starch grades: ADA, SOS1, and SOS3 prepared from emulsions stabilized with polysaccharides only (SD-ADA_0_100_, SD-SOS1_0_100_, SD-SOS3_0_100_), or polymer/polysaccharides 80/20 (SD-ADA_80_20_, SD-SOS1_80_20_, and SD-SOS3_80_20_).

**Figure 7 pharmaceutics-12-00571-f007:**
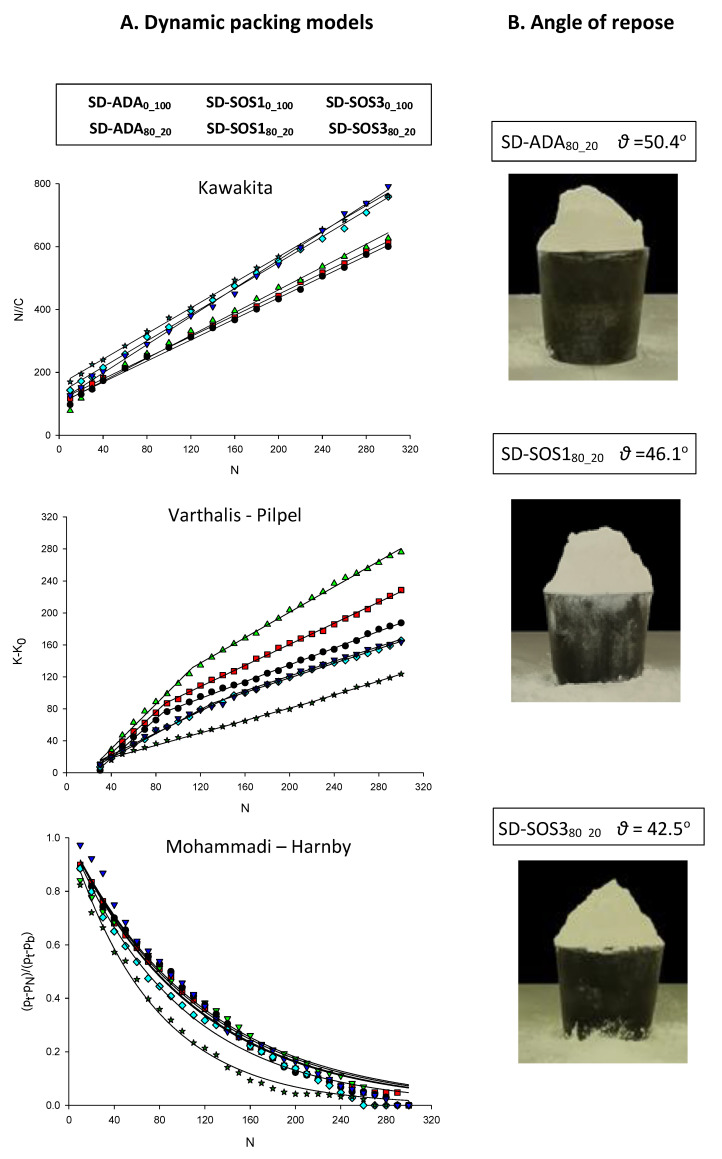
(**A**) Plots of densification functions vs. tapping number (N) according to Kawakita, Varthalis–Pilpel, and Mohammadi–Harnby equations. (**B**) Images of conical heaps of EO spray-dried products encapsulated in polymer/polysaccharides 80/20 wall (products with a polysaccharides wall only could not form a cone).

**Figure 8 pharmaceutics-12-00571-f008:**
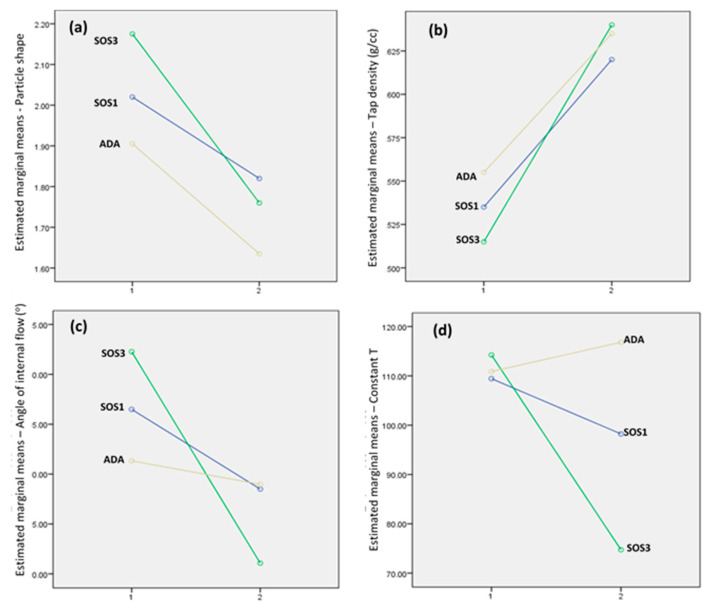
Statistical interaction plots of the effects of starch type and microencapsulating wall on: (**a**) particle shape, (**b**) tapped density, (**c**) angle of internal flow (Varthalis–Pilpel equation), and (**d**) constant T (Mohammadi–Harnby equation). On the x-axis, 1 represents a polysaccharide wall (PSC) and 2 represents a polymer/polysaccharide wall (PLM/PSC 80/20). Brown lines represent ADA, blue lines represent SOS1, and green lines represent SOS3 starch grades.

**Figure 9 pharmaceutics-12-00571-f009:**
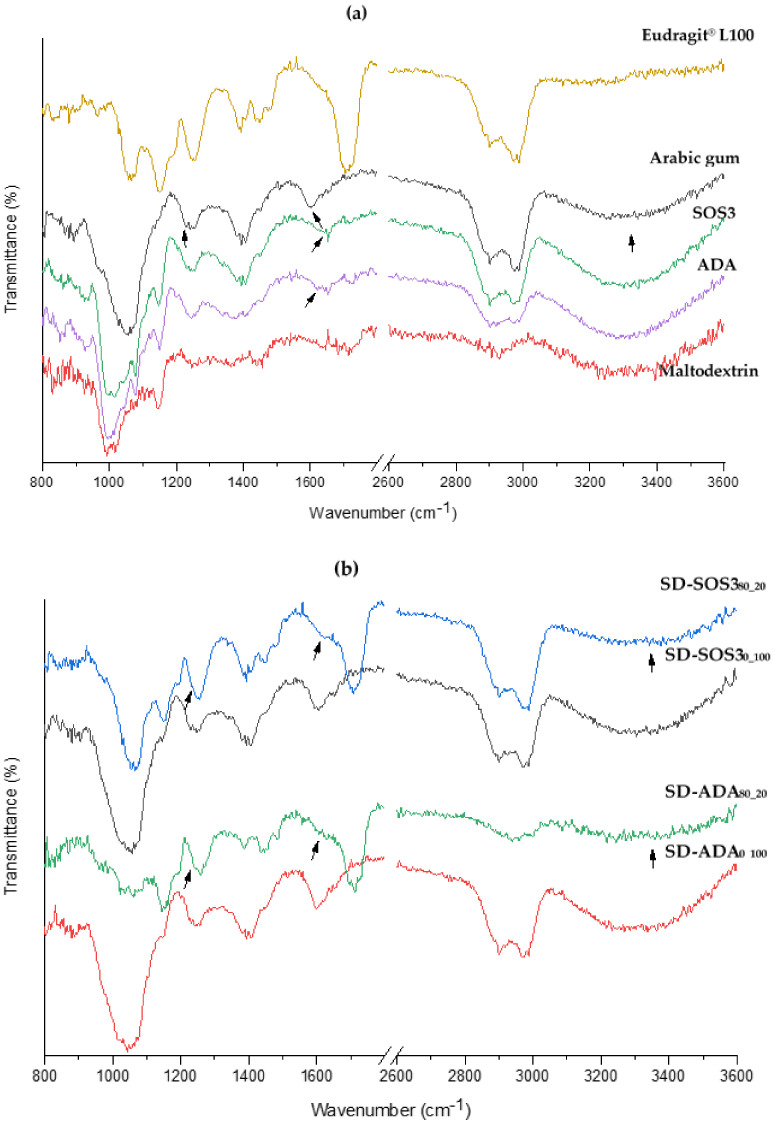
FTIR spectra of encapsulating wall materials (**a**) and spray-dried products (**b**) prepared from emulsions stabilized with polysaccharides only (subscript _0_100_) or polymer/polysaccharides 80/20 (subscript _80_20_) with modified starches ADA (SD-ADA_0_100_, SD-ADA_80_20_) and SOS3 (SD-SOS3_0-100_, SD-SOS3_80_20_).

**Figure 10 pharmaceutics-12-00571-f010:**
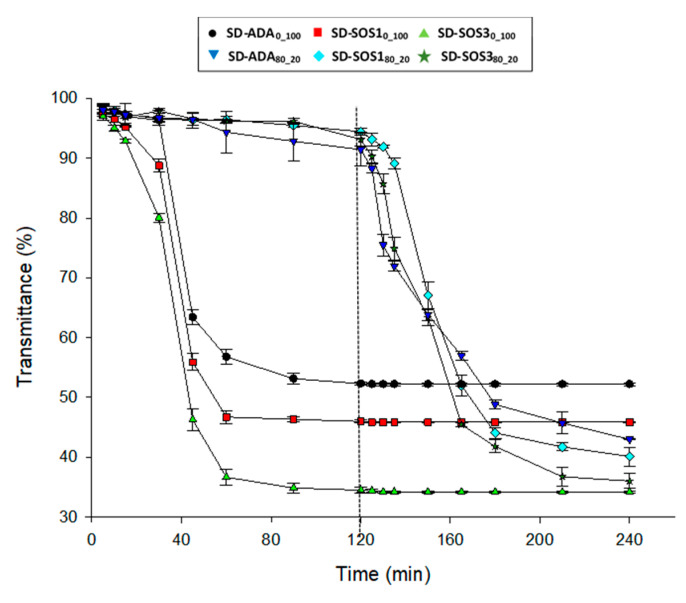
Transmittance of reconstitution medium vs. time for the spray-dried products (dotted line denotes pH change of the dissolution medium from 1.2 to 6.8).

**Figure 11 pharmaceutics-12-00571-f011:**
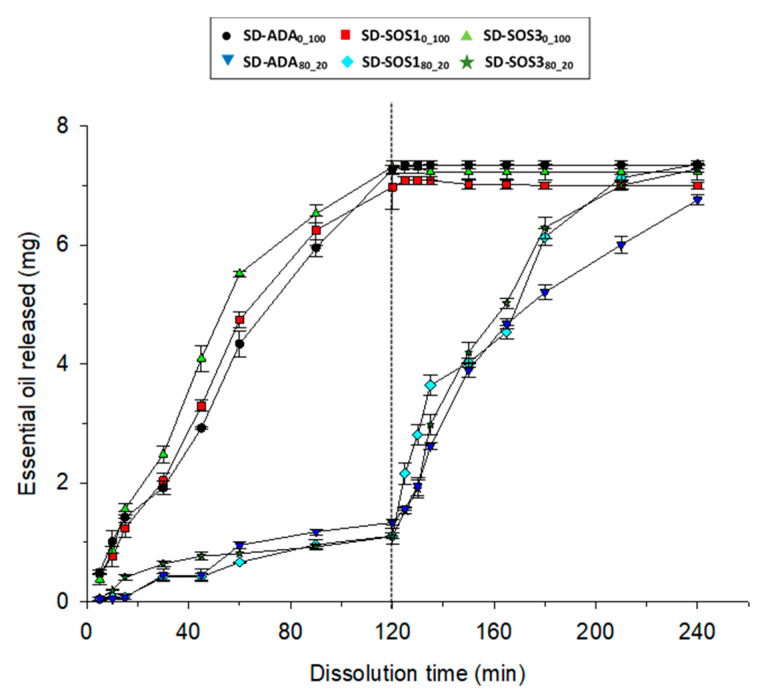
Release of oregano essential oil vs. time for the experimental spray-dried products (dotted line denotes pH change in the dissolution medium from 1.2 to 6.8).

**Table 1 pharmaceutics-12-00571-t001:** Coding of feed emulsions and corresponding spray-dried essential oil (EO) products and composition of the dispersed phase of the feed emulsions.

Code		Dispersed Phase (Microencapsulation Wall Plus EO)					PCS + PLM (% *w*/*w*)	Water % *w*/*w*
Feed Emulsion	Spray-Dried Product	Polysaccharides and EO (PCS) (% *w*/*w*)				Polymer (PLM) (% *w*/*w*)		
		EO	AG	MD	Starch Type	Eudragit^®^ L100		
ADA_0_100_	SD-ADA_0_100_	1.2	21.6	3.6	ADA/3.6	0	30	70
SOS1_0_100_	SD-SOS1_0_100_	1.2	21.6	3.6	SOS1/3.6	0	30	70
SOS3_0_100_	SD-SOS_30_100_	1.2	21.6	3.6	SOS3/3.6	0	30	70
ADA_80_20_	SD-ADA_80_20_	1.2	3.6	0.6	ADA/0.6	24	30	70
SOS1_80_20_	SD-SOS1_80_20_	1.2	3.6	0.6	SOS1/0.6	24	30	70
SOS3_80_20_	SD-SOS3_80_20_	1.2	3.6	0.6	SOS3/0.6	24	30	70

O: Oregano essential oil, AG: Arabic gum, MD: Maltodextrin, ADA: acetylated distarch adipate, PSC: polysaccharides, PLM: Eudragit^®^ L100, SOS1 and SOS3: high and low viscosity respectively sodium octenyl succinate starch.

**Table 2 pharmaceutics-12-00571-t002:** Spray-drying conditions.

Feed EO Emulsion	Inlet Temperature (°C)	Flow Rate (mL/min)	Air Flow (mL/min)
Emulsions with polysaccharides only	180	1.6	600
Emulsions with polymer	145	1.4	600

**Table 3 pharmaceutics-12-00571-t003:** Product yield, EO retention, moisture content, and physical properties of the spray-dried products.

Product	Yield (%)	Moisture Content (%) *	EO Retention (%)	Particle Density (g/cm^3^)	Particle Size (μm)			Span of Size Distribution	Shape Index	Surface Charge (nC) **
					d_10_	d_50_	d_90_			
SD-ADA_0_100_	25.0 ± 3.4	12.4 ± 0.11	67.5	1.249 ± 0.01	7.9	10. 7	18.4	0.99	1.96	−0.64 ± 0.08
SD-SOS1_0_100_	40.8 ± 3.1	12.1 ± 0.11	70.0	1.369 ± 0.01	8.5	10.5	15.9	0.70	2.04	−0.89 ± 0.07
SD-SOS3_0_100_	33.5 ± 2.2	12.0 ± 0.10	72.2	1.351 ± 0.01	7.6	9.2	15.1	0.82	2.17	−0.84 ± 0.11
SD-ADA_80_20_	29.9 ± 3.2	6.0 ± 0.08	73.5	1.414 ± 0.01	7.7	11.6	18.3	0.91	1.81	−0.04 ± 0.06
SD-SOS1_80_20_	49.9 ± 2.6	5.1 ± 0.08	73.5	1.396 ± 0.01	7.6	9.3	12.3	0.51	1.74	−0.14 ± 0.03
SD-SOS3_80_20_	39.6 ± 2.0	5.1 ± 0.06	72.8	1.383 ± 0.01	7.4	9.5	13.4	0.63	1.64	−0.02 ± 0.04

* Moisture content: SOS1 6.23%; SOS3 6.05%; ADA 9.07%; Maltodextrin: 4.86%; Arabic gum: 10.04%; Eudragit^®^ L100 3.41%; ** Surface charge of spray-dried Eudragit L100 particles 0.902 ± 0.19 nC.

**Table 4 pharmaceutics-12-00571-t004:** Densities and packing properties of the experimental spray-dried PSC and PLM/PSC powders.

Batch	ρ_b_ ^#^ (g/cm^3^)	ρ_t_ ^#^ (g/cm^3^)	Hausner Ratio	Carr’s Index (%)	Kawakita		Varthalis-Pilpel			Mohammadi-Harnby	
					A	R^2^	AIF (°)	INP (N) ^##^	R^2^	T	R^2^
SD-ADA_0_100_	0.28	0.56	2.00 ± 0.30	50.0 ± 2.2	0.54 ± 0.04	0.997	31.3 ± 1.6	27	0.997	110.9 ± 3.9	0.982
SD-SOS1_0_100_	0.28	0.54	1.95 ± 0.34	48.8 ± 3.1	0.57 ± 0.03	0.999	36.5 ± 1.7	33	0.998	109.4 ± 6.4	0.991
SD-SOS3_0_100_	0.27	0.51	1.92 ± 0.25	47.9 ± 2.6	0.55 ± 0.07	0.993	42.3 ± 1.7	36	0.998	114.2 ± 8.9	0.979
SD-ADA_80_20_	0.40	0.64	1.61 ± 0.13	37.9 ± 1.1	0.46 ± 0.01	0.998	29.0 ± 0.2	26	0.995	106.8 ± 1.8	0.979
SD-SOS1_80_20_	0.38	0.62	1.65 ± 0.17	39.5 ± 1.6	0.48 ± 0.02	0.998	28.5 ± 0.7	24	0.994	98.2 ± 2.4	0.984
SD-SOS3_80_20_	0.39	0.64	1.65 ± 0.07	39.4 ± 1.3	0.47 ± 0.01	0.998	21.1 ± 0.4	21	0.997	74.7 ± 1.9	0.992

^#^ SD ≤ 0.01; ^##^ SD ≤ 1; AIF: angle of internal flow; INP: inflection point.

**Table 5 pharmaceutics-12-00571-t005:** Angle of repose calculated by two methods: (a) cone geometry and (b) mean projected area obtained by image analysis.

Batch	Geometrical Method				Image Analysis			
	Hypotenuse (mm)	Height (mm)	cos*θ*	*θ* (^ο^)	Projected Area (mm^2^)	Height (mm)	cos*θ*	*θ* (^ο^)
SD-ADA_80_20_	35.9 ± 2.8	27.6 ± 3.6	0.637 ± 0.03	50.4 ± 2.18	804.3 ± 43.4	41.9 ± 1.9	0.550 ± 0.02	56.7 ± 1.45
SD-SOS1_80_20_	33.1 ± 2.1	23.8 ± 3.1	0.693 ± 0.01	46.1 ± 0.66	745.6 ± 13.5	39.8 ± 0.6	0.579 ± 0.01	54.6 ± 0.48
SD-SOS3_80_20_	31.2 ± 1.4	21.1 ± 1.9	0.737 ± 0.01	42.5 ± 0.67	644.8 ± 15.2	36.3 ± 0.7	0.634 ± 0.01	50.6 ± 0.66

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
