# Peer review of "Enteric Release Essential Oil Prepared by Co-Spray Drying Methacrylate/Polysaccharides—Influence of Starch Type"

_pharmaceutics, 2020, doi:10.3390/pharmaceutics12060571_

Round 1

Reviewer 1 Report

see attachment

Author Response

Reviewer 1:

Introduction

#1 The Manuscript deals with the use of polymers and polysaccharide compositions for the encapsulation of essential oils via spray drying and the effect of formulations on the powder handling properties. A broad variety of characterization methods was applied.

- We are very thankful for the extensive and thorough review of our work.

#2 Overall, the manuscript should be improved making the overall aim (and take-home message) of the manuscript clearer and by stating the contribution of each and every characterization method towards reaching this aim. The manuscript should undergo major revision and be reviewed again.

- The manuscript has been extensively revised according to reviewer’s suggestions.

Changes in the revised text are indicated by red colour.

The replies to the general (G) and specific (S) comments are given below in a point-by-point manner.

General Comments

G1 The Title of your manuscript is quiet long. You should consider to shorten it.

Response: Title has been shortened (lines 2-4).

G2 The key words should be adapted regarding the key objective of your study. Mention of specific evaluation methods (i.e. Kawakita, Varthalis-Plpel, Mohammadi-Harnby) should be avoided.

Response: Key words representing specific evaluation methods have been replaced (line 30).

G3 The abstract should be shortened and specified to the key objective and main findings presented in the manuscript

Response: Abstract has been shortened from 259 to 214 words and specified to the key objective and main findings (lines 13-29).

G4 The central theme of the manuscript is hardly accessible for the reader. The manuscript rather appears to be a sequence of results accessed by different analytical methods without putting different results in context to an overall research concept and/or superordinate aim.

Response: The overall research concept is to provide an enteric release product with good technological characteristics and potential for further development. The results of packing/ flowability prove this potential and the results of the measurement of physical properties (size, shape, electrostatics) explain this on the basis of fundamental particle and material properties. The spectroscopic studies and in-vitro release prove the enteric release characteristics of the products. The results of the work have been put in context by adding text in certain parts of the manuscript.

G5 Additionally, results are discussed little with respect to suitable literature

Response: Further discussion has been added in certain parts of the paper following the reviewer’s suggestions in the specific comments.

G6 Consider to change the format and layout of your graphs to achieve uniformity

Response: Format and layout of graphs has been changed to improve uniformity.

G7 Figures and tables should be relocated in the corresponding sections where the corresponding results are discussed

Response: Figures and Tables have been relocated appropriately, nearer to the relevant text.

Specific comments

S1 “Line 57: Change E coli to E. coli.”

Response: Change has been made (line 53).

S2 “Line 59: the term “carvacrol” cannot be linked to the described content. Is it the active substance in the oregon essential oil? If so, you should make this clear to the reader.”

Response: Carvacrol is the main constituent of oregano essential oil. Text has been added (line 55).

S3: “Line 74: spray drying is not necessarily a continuous process.”

Response: ‘continuous’ has been removed (line 71).

S4: “Line 78: What does “the same type equipment” stands for. Please be more specific.”

Response: ‘Same type of equipment’ has been further explained (line 75).

S5: “Line 106: The subheading 2.2.1. Characterization of starches is not deemed necessary, as this chapter only contains subchapter 2.2.1.1. viscosity. One subheading with a suitable caption should be sufficient.”

Response: One subheading 2.2.1. is used in the revised (line 104).

S6: “Line 108: Here it is described that the feed emulsions had a starch concentration of 3.7% w/w. In Table 1 (and the following results) the starch concentration amount to 3.6 % w/w. Please clarify this misleading information throughout the manuscript.”

Response: Thank you for the comment. The concentration value has been corrected to 3.6% w/w (line 105).

S7: “Line 109: The “)” seems to be wrongly located and should be placed after “emulsions”.”

Response: The “)” was relocated as suggested (line 106).

S8: “Line 110: Please delete “-“ between 20 and mL.”

Response: The “-” was deleted (line 107).

S9: “Line 113: If the equations for calculating the shear stress and shear rate are necessary for the manuscript, please introduce them as done e.g. in line135.”

Response: Equations have been introduced in separate lines 112, 113 of revised.

S10: “  Line 112-113: Please label the shear stress and the shear rate with the commonly applied symbols (σ and , respectively).”

Response: Shear stress and shear rate symbols have been replaced with σ and   as suggested (lines 109-113).

S11: “Line 114: Was the viscosity of the slopes of the linear parts of the graph estimated or calculated?”

Response: The slopes were estimated from best fitting regression lines.

S12: “Line 118: How was the EO encapsulated?”

Response: EO was encapsulated in the polysaccharide wall forming oil in water emulsion as described in lines 119-121. The Eudragit L100 dispersion was added to the EO emulsion and formed the final composite wall.

S13 “Line 118: What does “MS” stand for?”

Response: MS stands for modified starch and is explained in (line 92).

S14: “Line 120: Can you please specify the applied Ultra-Turrax. There a several types available. Please add information on the tip speed/circumferential velocity applied in the shearing gap.”

Response: The Ultra-Turrax model is TP18/10S5, IKA, Germany the circumferential velocity 19.95 m/s (line 120).

S15: “Line 125-127: Please comment on the selection of 30 % w/w dispersed phase in the feed based and stability, viscosity and minimal losses during drying. Was this evaluated in a prior study?”

Response: Yes, it was decided from preliminary trials

S16: “Line 129-130: Please subscribe 0_100 and 80_20 according to table 1. What do A,B, and C stand for? Do these letters reappear in the text?”

Response: We apologize for the confusion. A, B and C were replaced by “ADA”, “SOS1” and “SOS3”, as in Table 1 (lines 129, 130 of revised).

S17: “Line 132: please comment whether such low contents of EO are relevant for processing of a final dosage form.”

Response: The prepared formulations had 4% EO content. The dose to sick animals is 12.5 mg/kg and for a calf of 1-2 weeks old the weight is about 100-200 kg. This corresponds to a dose of 31.25-62.5 g which can be easily administered with food.  

S18: “Line 133: Please replace “%PLM content” by relative PLM content.”

Response: It has been replaced (line 133).

S19: “Line 133: Please replace “Equations” by “Equation”.”

Response: ‘Equations’ is correct because it refers to two equations, (3) and (4) (line 133).

S20: “Line 138-141: What to the “%” refer to? Mass or volume?”

Response: The “%” represents mass (Table 1).

S21: “Line 142-148: Please specify on the droplet size measurements. How many droplets where considered and what measure of size was used for describing the droplet sizes? How was the span of the droplet size distributions calculated?”

Response: About 600 droplets were analysed and size was expressed as equivalent circle diameter. The span of the droplet size distribution was defined as Span = (d90-d10)/d50. Explanatory text has been added in the revised (lines 145-150).

S22: “Line 150: Delete “quantity of” and “-“between “100” and “mL”.”

Response: Change has been applied (line 156).

S23: “Line 152: What does “173 °C” refer to?”

Response: There is a mistake, the values is 176 oC” and refers to decomposition (line 159).

S24: “Line 153: Replace “weight” by “mass”.”

Response: It has been replaced (line 159).

S25: “Line 154: Replace “%” by “relative”.”

Response: It has been replaced (line 160).

S26: “Line 158: May it be the case that denominator and numerator need to be exchanged?”

Response: Thank you for finding the mistake. Equation has been corrected (line 164).

S27: “Line 159, Table 2: Please give information about the mass/volume flow of the feed (i.e. spray rate) rather than the relative pump speed. Was the air flow parallel or counter-current to the spray feed? What does the temperature refer to (inlet or outlet)? As significantly different temperatures were applied, can it be excluded that this difference in process parameters influenced the product properties rather than differences in formulation parameters?”

Response:

-Information on feed rate in terms of volume is provided in Table 2.

-Different temperatures were applied for spray drying emulsions with polysaccharides only (PSC) and for emulsions with polysaccharides/Eudragit (PSC/PLM). PSC emulsions could not be spray-dried at temperatures lower that 180 oC due to sticking and inadequate drying, whereas emulsions with polysaccharides/Eudragit could not be spray-dried at such high temperatures due to decomposition. Therefore, optimal inlet temperatures were used in each case. The effect of the difference in inlet temperature is not excluded (lines 380-384).

- Air flow was parallel to the spray feed (line 156). The temperature refers to inlet air (Table 2).

-In the present experiments Pe and particle formation should be mainly influenced by the drying time or the evaporation rate which was faster in the case of PSC products dried at higher inlet temperature (lines 379-383). However, we believe that the elimination of cavities is resulting from the presence of Eudragit in the wall as explained in lines 383-386.

S28: “Line 161: Please comment on how many particles were measured, which measure for particle size were used and how powder samples were dispersed in order to measure particle sizes properly.”

Response: The requested information is provided in lines 169-172. We have revised this part of the text and added the requested information (lines 169-172).

S29: “Line 164: Please format the equation properly. Where does this equation come from?”

Response: Equation has been formatted (line 174). The equation originates from Hausner, H.H., 1966. Characterisation of the powder particle shape. Planseeber. Pulvermetall. 14, 75–84) and further applications can be found in powder technology textbooks (e.g. Enlargement and Compaction of Particulate Solids by Stanley-Wood Butterwoths).

S30: “Line 166: Please add “qualitatively” after examined.”

Response: “qualitatively” has been added (line 175).

S31: “Line 167: What does “20 kV” refer to?”

Response: It refers to the electron beam voltage in the SEM (line 176).

S32: “Line 187-188: Please format the equations properly.”

Response: Equations have been formatted (lines 198, 199).

S33: “Line 196: Please format the equation properly.”

Response: Equation has been formatted (line 212).

S34: “Line 196 – 197: Specify the values of “a” and “b” and where you get them for your powder from.”

Response: Values of constants ‘a’ and ‘b’ are derived from fitting regression straight lines to Kawakita plots from the slope and the intercept values.

S35: “Line 199: Please format the equation properly.”

Response: Equation has been formatted (line 215 and 220).

S36: “Line 191 – 202: Description of the application of the different models is not sufficient for understanding of the procedure. Please make the experimental setup or rather the calculation of the different parameters more clear. It should be additionally noted how the parameters/outcome of the different models can be related to the microstructure or flowability of the powder.”

Response: The part of the manuscript has been re-written (lines 201-226) to give a better description of the model and of the calculation of the important parameters.

S37: “Line 209: Please specify what the hypotenuse is in this case.”

Response: The hypotenuse is specified in the revised (lines 233-234).

S38: “Line 225: Please comment on changing the operation mode in comparison to stability testing. How was the reconstitution of the powders assessed specifically (indicator for re-emulsification property)?”

Response: The instrument provides measurements of light scattered by the dispersed droplets and non-scattered light transmitted through the continuous medium (external phase of the dispersion). Reconstitution is assessed from the drop of transmittance relative to that of distilled water.

S39: “Line 234: Please specify the device used for UV spectroscopy.”

Response: The device of UV spectroscopy has been specified in the revised (lines 259-260).

S40: “Line 261, Figure 1: Is it necessary to depict this graph or would it be sufficient to just describe the main findings of the corresponding publication (i.e. different interfacial tensions for the different materials)? Additionally, there are 4 different graphs in figure 1 but the corresponding legend only contains 3. Please adapt to maintain comprehensibility.”

Response: Figure 2 has been corrected to show 3 graphs that are of interest to our work.

S41: “Line 268: Rewrite to “...a linear increase of the shear stress with increasing shear rate, indication Newtonian behaviour”.”

Response: Text has been re-written as suggested (lines 293-294).

S42: “Line 273: Was the influence of the different viscosities on droplet formation investigated? If not rewrite in order to clarify that the stated statement is an assumption.”

Response: Text has been added (line 299)

S43: “Line 273-274: Can you specify the influence on the molecular arrangement in the encapsulating wall?”

Response: Yes. This will assist intermolecular packing of the PSC with PLM molecules (lines 300-301).

S44: “Line 279: Specify the heading (e.g. Characterization on feed emulsions)”

Response: It has been done (line 305).

S45: “Line 280-291: Please comment on reproducibility of feed emulsion production, standard deviations of droplet size measurement characteristics and how differences in droplet size may be explained or why are droplet sizes are important with regard to the superordinate aim of the manuscript.”

Response: - Droplet size measurements were derived from a large number (about 600) droplets. Standard deviations between different measurement fields are not meaningful. The span values are presented to show the spread of the distribution.

-The importance of droplet size in relation to the aim of the manuscript is to show that the emulsions were stable and the whole two-step process robust.

S46: “Line 288, Figure 4: Pictures of feed emulsions are partly of poor quality. Please adapt in order to facilitate better comparability of droplet sizes. Additionally, a scale bar should be included.”

Response: The quality of the pictures has improved and scale bars have been added (Figure 4).

S47: “Line 299, Figure 5: Please adapt labelling of axis (“label” [unit]] and legend (concerning different colours) to facilitate comprehensibility.”

Response: In Figure 5 of revised X-axis has been labelled and the different color curves explained (lines 328-329).

S48: “Line 311-312: Yields appear to be really low. May it be the case, that process parameters were not adapted properly to the investigated formulations or are such low yields normal during such spray drying processes?”

Response: Spray-drying conditions were optimal for the B-191, Büchi lab-scale spray dryer. Such yields are expected for this of equipment and polysaccharide based encapsulating materials.

S49: “Line 313-315: This sentence is not comprehensible. Where do the MC% values for the polysaccharides come from?”

Response: The method of determination of the MC% for all experimental spray-dried powders is described in section 2.2.5 and corresponding values are presented in Table 3 (lines 339-340).

S50: “Line 320-322: Yield aligns with the viscosity of the corresponding feed solutions. How can this be explained? Did you actually observe a less deformable encapsulating wall? How may this influence yield exactly?”

Response: Measurement of the mechanical properties of the encapsulating wall is beyond the scope of this study. From the available results however, since viscosity or reduced fluidity is associated with contact surface area and adherence, it is mentioned as a possible reason for the higher yield of SOS1. 

S51: “Line 323-328: Around 30 % of the EO get lost during drying. Can this be related to a separation and classification of the different materials during the process or rather to a degradation of the oil? How does the temperature influence EO retention? Described values for the drying temperature do not match the materials and methods section. Low yields in combination with low EO retention picture the spray drying process as not suitable for drying of EO. Comment on that. Why is that the case and how can that be improved?”

Response: Thank you for giving us the opportunity to discuss this point

The loss of EO is due to evaporation during spray drying.

There is no degradation, only change in the consistency due to greater loss of the more volatile ingredients in the case of oregano oil p-Cymene, γ-Terpinene, other terpenes (Partheniadis et al. 2019).

The loss of EO increases with oil concentration in the feed emulsions (Partheniadis et al 2019) and the volatile ingredients. This ratio of encapsulated agent/wall material has also been shown to affect the overall spray drying of essential oils but also of drugs (Czyz et al., 2020 EJPS).

The EO loss also depends on the glass transition of the matrix formers since this determines the drying temperature.

We are currently investigating different materials to reduce the loss and make spray drying a better choice for producing EO in powder form.

Drying temperature has been corrected to match the value in the Materials and methods (line 352).

S52: “Line 327: May the difference in particle density between PSC and PLM/PSC rather be ascribed to differences in densities of applied materials rather than to the structure of the powder particles? A comparison between the different formulations seems not to be suitable here.”

Response: The differences are significantly different and cannot be merely explained on the basis of materials.

S53: “Line 342-348: Please make the application of the Pe-number to your powder particles more comprehensible.”

Response: Text has been revised, lines 379-383.

S54: “Line 351-352: Did you actually observe that Eudragit results in a more rigid encapsulation wall? If not, please refer to corresponding literature supporting your assumption.”

Response: Reference has been added (Jose C. Gutierrez-Rocca & James W. Mcginity 1993) (line 387).

S55: “Line 353, Figure 6: Please consider making the differentiation between PSC and PLM/PSC samples more clear. What do the white numbers at the bottom of the images stand for (e.g. 8101 1445 SEI top right)? If these numbers are not necessary, please delete them from the images).”

Response: Unfortunately, we cannot interfere with the pictures. This is how it is delivered from the scanning electron microscope.

S56: “Line 361-363: May differences in electrostatic charges explain also differences in process yield? Please comment on the standard deviations of the corresponding values.”

Response: The spray-dryer was earthed.  Comment on the standard deviations has been added (line 398-399).

S57: “Line 367: Change “should” to “may”.”

Response: Change has been made (line 397).

S58: “Line 373: Please change “pattern” to “trend”.”

Response: Change has been made (line 404).

S59: “Line 382, table 4: The table is too big and exceeds the available width. Consider to put the dynamic models in a separate table and put this in the corresponding section. Please change “g/cc” to g/cm3.”

Response: We prefer to leave Table 4 as it is gives the parameters determined by tapping altogether. However, if the reviewer insists the change will be made.

S60: “Line 392-393: Why is the evaluation of the whole packing process even of interest in the context of the superordinate aim of the manuscript?”

Response: Because it differentiates the packing ability of the different spray- dried products which is a very important attribute that determines their usability.

S61: “Line 398: Do the applied dynamic models actually consider the whole packing process? If so, please comment how and what can be extracted from the different models concerning the packing process for a better comprehensibility.”

Response: They applied dynamic models do not consider the entire packing process but they derived from a greater number of volumetric measurements obtained every 10 taps. Text has been changed (lines 420-421 and 426-4267). The info that can be extracted is described in lines 201-226.

S62: “Line 399, Figure 7: Why are these figures located here and not in the corresponding sections. Why are results from the dynamic models put together in one figure with the angle of repose? Please add units for the axis labels.”

Response: Figure 7 has been moved near the point of discussion. The graphical presentation of the dynamic models is given in the same Figure to facilitate comparison. There are no units assigned to the variables in the axes because they are dimensionless.

S63: “Line 404: Applicability of the different models to your powder should be explained more comprehensible (either here or in the materials and methods section).”

Response: Explained in the materials and methods in lines 201-206.

S64: “Line 408: How can the parameter “a” be extracted from the graph and why can it be referred to the packing ability?”

Response: Please refer to the response to comment S36.

S65: “Line 410-414: Are the described phenomena actually observable from the corresponding graph? It seems that a reference is necessary here.”

Response: They are observable. Reference could not be found so explanatory sentence was added.

S66: “Line 417: Less frictional forces for which powders?”

Response: It implies the PLM/PSC powders (text has been added in line 450).

S67: “Line 420-421: How is the parameter “T” related to the microstructure of the powder bed?”

Response: Lower T values reflect a structure consisting of rigid single particles or rigid particle agglomerates, i.e. PLM/PSC particles due to the presence of Eudragit L100 in the encapsulating wall, whereas higher T values reflect weakly agglomerated particles, i.e. PSC particles. Text has been added (lines 454-457).

S68: “Line 404-424: How is the outcome of the dynamic models different from the other indices (Hausner and Carr)?”

Response: Hausner and Carr represent only two states of powder bed, initial and final, without describing the entire process. Therefore, their discriminating ability for the present experimental powders with similar particle size is inadequate. For this reason, the dynamic packing behaviour of the powders was evaluated by recording volumetric changes every 10 taps up to 300 and processing the results using known packing models (lines 201-206).

S69: “Line 425-435: Please describe the statistical analysis and its outcome in a more comprehensible manner and relocate it to the methods section. Locate the corresponding figure here, in the discussion of the results, and describe how the figure can be read and what the outcome indicates precisely.”

Response: Text describing the experimental design and reference have been added (lines 467-471). Adding more information would make the text unnecessary lengthy since the methods can be found in standard textbooks. Text has been added to explain better what the outcome indicates (lines 478-482). Figure has been relocated as suggested.

S70: “Line 445: What happened instead for the PCS powders?”

Response: PSC powders did not form conical heap and hence angle of repose could not be estimated (lines 496-497).

S71: “Line 450: Please improve comprehensibility of the graphs (legend, axis labelling) and align graphs.”

Response: Information has been added to the graphs

S72: “Line 460-463: This is not really signified by the shown results, as you did not specifically investigate the molecular structure of the encapsulating wall. The described phenomena are at most indicated by your results if discussed with suitable literature.”

Response: Statement has been modified (lines 507-508).  

S73: “Line 375-466: Please discuss the powder packing evaluation by the different methods in a more comparable manner.”

Response: Discussion has been added (lines 428-434, 468-472, 478-482).

S74: “Line 467-493: A final conclusion of the presented results with respect to previously shown results and/or the superordinate aim of your study is missing.”

Response:

S75: “Line 499-521: Please explain how the transmittance can be interpreted regarding emulsion reconstitution.”

Response: Decrease of transmittance is due to reconstitution of emulsion, i.e. due to the presence of oil droplets reducing light transmittance along the test tube.

S76: “Line 504: Change “is” to “are”.”

Response: It has been changed (line 554).

S77: “Line 505: Delete “(…)”. This should be clarified by the figure itself.”

Response: It has been removed (line 555).

S78: “Line 508-510: May these differences also be caused by the differences in EO retention described before?”

Response: Text has been added (lines 560-561).

S79: “Line 530: Please make clear what the number 155.22 stand for (e.g. by including a variable and a unit).”

Response: It has been done (lines 581 – 582).

S80: “Line 534: Change the “,” after “min” to a “.”..”

Response: It has been done (line 585).

S81: “Line 537: Please comment on the effect of 15% release under gastric condition on gastric irritation potential and by that regarding the success of encapsulation.”

Response: This may be due to the presence of non-encapsulated EO on the surface of the particles.

S82: “Line 546, Figure 11: Depicting the relative release of EO against time would be more comprehensible and would facilitate better comparability of different formulations.”

Response: We prefer to leave show the actual released amount since this shows better the differences between the spray dried products. Also, it shows agreement between the released EO of about 7.5 mg out of 10 mg nominal amount added (nominal) and the retention.

Reviewer 2 Report

L117  What is MS?

L123  NH3   3 should be subscript

At Spray drying  I could not find liquid flow rate

L225 [5,30]  please add space after , .

232 2M  M changes to N. N is better

Figure 4,  Please add scale bar.

Table 3,   cc to cm3

L539  Fig 4  add Fig.   period

Author Response

Reviewer 2:

We are very thankful to the Reviewer for constructive comments and suggestions. The questions raised by the reviewer (Q) have been responded below by a point-by-point manner. Changes in the revised text are indicated by red color.

Q1. “L117 What is MS?”

Response: MS stands for modified starch and is explained in (line 92).

Q2. “L123 NH3 3 should be subscript”

Response: Thank you for indicating this mistypingr. “3” in NH3 has been subscripted (line 123).

Q3. “At Spray drying I could not find liquid flow rate”

Response: Flow rate information has been added in Table 2.

Q4. “L225 [5,30] please add space after , .”

Response: Space has been placed after the “,” (line 250).

Q5. “232 2M M changes to N. N is better”

Response: “M” has been changed to “N”, as suggested (lines 256-257).

Q6. “Figure 4, Please add scale bar.”

Response: Scale bar has been added to Figure 4.

Q7. “Table 3, cc to cm3”

Response: “cc” has been changed to “cm3” in Table 3.

Q8. “L539 Fig 4 add Fig.  period”

Response: Period “.” Has been added after “Fig” (line 590).

Round 2

Reviewer 1 Report

General Comments:

  • Generally, the scientific writing should be improved (e.g. space between value and the corresponding unit (also for percentages), axis labelling with corresponding units, even when depicted values are dimensionless ([-]).
  • Consider to change the format and layout of your graphs to achieve uniformity. Instead of using graphs as obtained from the individual analytical devices, graphs may be prepared based on raw data in a suitable data analysis/graphing software. However, this point is not crucial for an acceptance of the manuscript.

Specific Comments:

Materials and Methods

  • Line 135: It is not comprehensible where the values (28.8, 1.33, 8) in equation 8 come from. Please, either specify the origin of these values or (better) adapt the equations in a more general manner by replacing the individual values by a general variable, which is linked to the specific value in the text.
  • Line 138-141: The expression “polysaccharides” in table 1 (line 2, columns 3-6) is misleading as a header for the subjacent cells, as the corresponding substances are not only polysaccharides (e.g. EO, AG). What does MA stand for?
  • Line 148-150: Please subscribe “10”,”50” and “90” for the different particle size fractions in the text and in equation (5).
  • Line 174: If there is a suitable literature for equation (8) (Hausner, H. H., 1966), it should be cited.
  • Line 205-207: The statement “These models have been found to provide useful information on the behaviour of fine pharmaceutical powders (…)” should be supported by corresponding literature.
  • Line 209: Please put a multiplication symbol between “a” and “b” in equation (11).
  • Line 211: If “a” and “b” are derived from fitting regression straigt lines to Kawakita plots, this procedure should be explained in the manuscript.
  • Line212: Please subsribe “300” after “V”.
  • Line 216: What does “K0 stands for in equation (13)? Please delete the brackets around “AIF” and put a multiplication sign between “AIF and “N”.
  • Line 221: Please subscribe intended subscripts in equation (14) and introduce the corresponding variable in the text.
  • Line 224: What does “rN” stands for in equation (15)? Please add a multiplication sign after “D”.

Results and Discussion:

  • Line 293 & 296: Please adapt the concentrations of modified starches to 3.6 % w/w.
  • Line 326: What do the coloured numbers on the right y-axis stand for? If these should indicate the measuring time, please put them in a legend and add a corresponding unit.
  • Line 349-350: Please make the connection of higher yields with a higher viscosity and a less deformable encapsulating wall clear to the reader and support your hypothesis by suitable citations.
  • Line 357-359: Please subscribe “10”,”50” and “90” for the different particle sizes. Listing “Span” under “Particle size (µm)” is misleading, as the span of a particle size distribution is dimensionless. Please add another “*” after “Surface charge (nC)”.
  • Line 369-370: Consider erasing unimportant information from figure 6. Even if these pictures were delivered from the SEM as they are depicted, it is possible to edit them (e.g. cropping and manual adding of scale bar) in a suitable image editing program.
  • Line 430: Please subsribe”50” after “d”.
  • Line 432: Delete “particle” after morphological.
  • Line 461-462: Please label axis of figure 7 with corresponding units. If depicted variables are dimensionless indicate this e.g. by adding [-].
  • Line 483: Please add a legend in figure 8 for the differently coloured graphs.
  • Line 513: The footer of table 5 (“*”) cannot be allocated (second “*” is missing).

Conclusion:

  • The conclusion is really short with respect to the presented results. Please consider an amplification. Additionally, you should consider giving an outlook with respect to the presented findings.

Author Response

MDPI Pharmaceutics

Manuscript ID: pharmaceutics-809231

We are thankful to the reviewer for his comments and suggestions

The replies to the specific (S) comments are given in a point-by-point manner.

Τhe requested changes are marked in the revised text by red colour.

Reviewer 1 – Round 2:

Replies to general comments (G)

G1 Generally, the scientific writing should be improved (e.g. space between value and the corresponding unit (also for percentages), axis labelling with corresponding units, even when depicted values are dimensionless ([-]).

- Spaces between the values and corresponding units were placed as suggested. Graphs have also been revised for axis labelling.

G2 Consider to change the format and layout of your graphs to achieve uniformity. Instead of using graphs as obtained from the individual analytical devices, graphs may be prepared based on raw data in a suitable data analysis/graphing software. However, this point is not crucial for an acceptance of the manuscript.

- Since the graphs are quite intelligible and descriptive we prefer not to change them to keep the originality and because information may be lost or altered during transfer to a different format.

 Replies to specific comments (S)

Materials and Methods

 S1 Line 135: It is not comprehensible where the values (28.8, 1.33, 8) in equation 8 come from. Please, either specify the origin of these values or (better) adapt the equations in a more general manner by replacing the individual values by a general variable, which is linked to the specific value in the text.”

Response: Equations (3) and (4) were revised as suggested (lines 132-139).

S2 Line 138-141: The expression “polysaccharides” in table 1 (line 2, columns 3-6) is misleading as a header for the subjacent cells, as the corresponding substances are not only polysaccharides (e.g. EO, AG). What does MA stand for?”

Response: The expression “polysaccharides” in Table 1 has been revised. “MA” has been corrected to “MD” which stands for maltodextrin (stated in line 92).

S3: Line 148-150: Please subscribe “10”,”50” and “90” for the different particle size fractions in the text and in equation (5).

Response: “10”, “50” and “90” have been subscripted in the text and equation (5) (lines 151 and 153).

S4: Line 174: If there is a suitable literature for equation (8) (Hausner, H. H., 1966), it should be cited.

Response: Reference for equation (8) has been added (ref 24, line 175).

S5: Line 205-207: The statement “These models have been found to provide useful information on the behaviour of fine pharmaceutical powders (…)” should be supported by corresponding literature.”

Response: Reference no 27 has been added (line 210).

S6: Line 209: Please put a multiplication symbol between “a” and “b” in equation (11).”

Response: A multiplication symbol (×) has been added between “a” and “b” in equation (11).

S7: Line 211: If “a” and “b” are derived from fitting regression straigt lines to Kawakita plots, this procedure should be explained in the manuscript.”

Response: Explanatory text has been added (line 217).

S8: Line 212: Please subsribe “300” after “V”.

Response: “300” after “V” was subscribed (line 215).

S9: Line 216: What does “K0” stands for in equation (13)? Please delete the brackets around “AIF” and put a multiplication sign between “AIF” and “N”.”

Response: The brackets around AIF were deleted and a multiplication sign (×) was added in equation (13). Ko stands for the intercept on the ordinate of the straight line when plotting K vs N. Explanatory text has been added (lines 222-223).

S10: Line 221: Please subscribe intended subscripts in equation (14) and introduce the corresponding variable in the text.

Response: Equation (14) has been revised as suggested and variables explained (line 230).

S11: Line 224: What does “rN” stands for in equation (15)? Please add a multiplication sign after “D”.”

Response: “ρN” stands for the powder bed density after N taps. Appropriate text has been added in the revised manuscript (line 231). A multiplication sign (×) has been added between “D” and “exp” in equation (15) (line 230).

Results and Discussion:

S12: Line 293 & 296: Please adapt the concentrations of modified starches to 3.6 % w/w.

Response: The values of the concentration of modified starches have been adapted to 3.6 % w/w (lines 300 and 303).

S13: Line 326: What do the coloured numbers on the right y-axis stand for? If these should indicate the measuring time, please put them in a legend and add a corresponding unit.

Response: Coloured numbers are explained in the legend of Figure 5.

S14: Line 349-350: Please make the connection of higher yields with a higher viscosity and a less deformable encapsulating wall clear to the reader and support your hypothesis by suitable citations.

Response: Text has been added (lines 356-359) and supplemented by new reference no 38.

S15: Line 357-359: Please subscribe “10”,”50” and “90” for the different particle sizes. Listing “Span” under “Particle size (µm)” is misleading, as the span of a particle size distribution is dimensionless. Please add another “*” after “Surface charge (nC)”.

Response: Table 3 has been revised as suggested.

S16: Line 369-370: Consider erasing unimportant information from figure 6. Even if these pictures were delivered from the SEM as they are depicted, it is possible to edit them (e.g. cropping and manual adding of scale bar) in a suitable image editing program.

Response:

Unimportant information has been removed from Figure 6.

S17: Line 430: Please subsribe”50” after “d”.”

Response: “50” after “d” was subscribed  (line 438).

S18: Line 432: Delete “particle” after morphological.

Response: “particle” has been deleted (line 440).

S19: Line 461-462: Please label axis of figure 7 with corresponding units. If depicted variables are dimensionless indicate this e.g. by adding [-].”

Response: Figure 7 has been revised as suggested.

S20: Line 483: Please add a legend in figure 8 for the differently coloured graphs.”

Response: Insert labels have been added next to the curves individually.

S21: Line 513: The footer of table 5 (“*”) cannot be allocated (second “*” is missing).”

Response: Footer has been removed and the information was moved to the Methods section, lines 236, 237.

Conclusion:

S21: “The conclusion is really short with respect to the presented results. Please consider an amplification. Additionally, you should consider giving an outlook with respect to the presented findings.”

Response: The Conclusions section has been enlarged and authors’ outlook with respect to the present findings added (lines 616-619, 626-628).